# Systems Biology Approach Identifies Prognostic Signatures of Poor Overall Survival and Guides the Prioritization of Novel BET-CHK1 Combination Therapy for Osteosarcoma

**DOI:** 10.3390/cancers12092426

**Published:** 2020-08-26

**Authors:** Pankita H. Pandya, Lijun Cheng, M. Reza Saadatzadeh, Khadijeh Bijangi-Vishehsaraei, Shan Tang, Anthony L. Sinn, Melissa A. Trowbridge, Kathryn L. Coy, Barbara J. Bailey, Courtney N. Young, Jixin Ding, Erika A. Dobrota, Savannah Dyer, Adily Elmi, Quinton Thompson, Farinaz Barghi, Jeremiah Shultz, Eric A. Albright, Harlan E. Shannon, Mary E. Murray, Mark S. Marshall, Michael J. Ferguson, Todd E. Bertrand, L. Daniel Wurtz, Sandeep Batra, Lang Li, Jamie L. Renbarger, Karen E. Pollok

**Affiliations:** 1Department of Pediatrics, Hematology/Oncology, Indiana University School of Medicine, Indianapolis, IN 46202, USA; phpandya@iu.edu (P.H.P.); msaadatz@iu.edu (M.R.S.); khbijang@iu.edu (K.B.-V.); memurray@iu.edu (M.E.M.); mmarsha@iu.edu (M.S.M.); micjferg@iu.edu (M.J.F.); batras@iu.edu (S.B.); 2Department of Pediatrics, Herman B Wells Center for Pediatric Research, Indiana University School of Medicine, Indianapolis, IN 46202, USA; bajbaile@iu.edu (B.J.B.); chemenwa@iupui.edu (C.N.Y.); jixinding@gmail.com (J.D.); erzimmer@iu.edu (E.A.D.); adilyelmi09@gmail.com (A.E.); qthomps@iu.edu (Q.T.); fbarghi@iu.edu (F.B.); jeremiahshultz13@gmail.com (J.S.); heshanno@iu.edu (H.E.S.); 3Department of Biomedical Informatics, College of Medicine, Ohio State University, Columbus, OH 43210, USA; Lijun.Cheng@osumc.edu (L.C.); Shan.Tang@osumc.edu (S.T.); 4In Vivo Therapeutics Core, Indiana University Melvin and Bren Simon Comprehensive Cancer Center, Indiana University School of Medicine, Indianapolis, IN 46202, USA; alsinn@iu.edu (A.L.S.); mtrowbri@iu.edu (M.A.T.); kathcoy@iu.edu (K.L.C.); savdyer@iu.edu (S.D.); 5Department of Medical and Molecular Genetics, Indiana University School of Medicine, Indianapolis, IN 46202, USA; 6Department of Pathology, Indiana University School of Medicine, Indianapolis, IN 46202, USA; eaalbrig@iu.edu; 7Department of Orthopedics Surgery, Indiana University School of Medicine, Indianapolis, IN 46202, USA; tod.bertrand@gmail.com (T.E.B.); dwurtz@iupui.edu (L.D.W.); 8Indiana Institute of Personalized Medicine, Indiana University School of Medicine, Indianapolis, IN 46202, USA; 9Department of Pharmacology and Toxicology, Indiana University School of Medicine, Indianapolis, IN 46202, USA

**Keywords:** osteosarcoma, precision genomics, molecular signature, biomarkers, MYC, RAD21, CHK1, BETs

## Abstract

Osteosarcoma (OS) patients exhibit poor overall survival, partly due to copy number variations (CNVs) resulting in dysregulated gene expression and therapeutic resistance. To identify actionable prognostic signatures of poor overall survival, we employed a systems biology approach using public databases to integrate CNVs, gene expression, and survival outcomes in pediatric, adolescent, and young adult OS patients. Chromosome 8 was a hotspot for poor prognostic signatures. The MYC-RAD21 copy number gain (8q24) correlated with increased gene expression and poor overall survival in 90% of the patients (*n* = 85). MYC and RAD21 play a role in replication-stress, which is a therapeutically actionable network. We prioritized replication-stress regulators, bromodomain and extra-terminal proteins (BETs), and CHK1, in order to test the hypothesis that the inhibition of BET + CHK1 in MYC-RAD21+ pediatric OS models would be efficacious and safe. We demonstrate that MYC-RAD21+ pediatric OS cell lines were sensitive to the inhibition of BET (BETi) and CHK1 (CHK1i) at clinically achievable concentrations. While the potentiation of CHK1i-mediated effects by BETi was BET-BRD4-dependent, MYC expression was BET-BRD4-independent. In MYC-RAD21+ pediatric OS xenografts, BETi + CHK1i significantly decreased tumor growth, increased survival, and was well tolerated. Therefore, targeting replication stress is a promising strategy to pursue as a therapeutic option for this devastating disease.

## 1. Introduction

High-grade osteosarcoma (OS) is the most common primary bone cancer in children, as well as in adolescents and young adults (AYA) [1,2]. The aggressiveness of this disease is highlighted by the fact that 40% of OS patients develop metastases over time, with 15–20% of OS patients already exhibiting metastases at initial diagnosis [3]. Although the etiology of OS is still unclear, studies have found that mesenchymal stem cells with partial commitment to the osteoblastic lineage acquire genetic mutations that contribute to the onset of OS [3]. Despite a decrease in childhood cancer-related mortality rates due to advancements in therapeutic approaches, the survival rates for pediatric and AYA OS patients have remained unchanged in the last 30 years [4]. In contrast to other pediatric sarcomas, OS lacks a specific oncogenic driver, which further complicates the identification of therapeutic targets [5]. Additionally, in the last 3–4 decades, early phase clinical trials have failed to show promise for the treatment of not only primary OS, but also the more aggressive (relapsed/refractory) form of the disease [6]. The rarity of this disease and small patient populations continue to pose a challenge for initiating new clinical trials for pediatric and AYA OS patients [7]. Therefore, it is critical to understand the molecular heterogeneity and identify biomarkers within the complex genetic landscape of OS, so that actionable targets can ultimately be tested in clinical trials [3].

A detailed understanding of the molecular mechanisms involved in OS pathogenesis is helping to delineate potential prognostic biomarkers as well as predictive biomarkers of therapeutic response in this disease [4,8]. A number of efforts have focused on the large-scale characterization of genetic, epigenetic, and transcriptional changes to identify novel targets or combinations of targets that are promising candidates for testing in clinical trials for pediatric and AYA OS [4]. OS is characterized by various copy number variation (CNV)-based genomic instabilities, such as copy number gains in chromosome arms 6p, 8q, and 17p, among many others [9,10]. In our previous study, we utilized publicly available datasets to integrate patient CNVs and survival outcomes with in-vitro drug response data associated with specific CNVs in pediatric sarcoma cell lines [11]. Studies by others have revealed OS-related CNVs in many potential oncogenes, such as *RUNX2*, *CDC5L*, *VEGFA*, *PIM1*, *E2F3*, *TWIST*, *MYC*, *PRIM1*, *CDK4*, *MDM2*, *COPS3*, *PMP22*, and *MAPK7* [12,13,14,15,16,17,18,19,20,21,22,23,24,25,26,27]. Tumor suppressors have also been reported in OS-related CNV studies and include deletions in *INK4A*, *ARF*, *INK4B*, *WWOX*, and *TP53* [13,28,29,30,31,32]. While these studies investigated correlations between CNVs or gene expression and patient outcomes [5,33,34,35,36,37], they did not comprehensively integrate genome-wide discovery with CNVs, gene expression, and patient outcomes. There is still a critical gap between linking candidate biomarkers of the therapeutic response to actionable targets, signaling pathways, and networks. Moreover, others have demonstrated that single-level analysis which focuses on one level of pathogenesis, such as the transcriptome or proteome, is not sufficient for the identification of novel therapeutic targets in OS [38]. In addition, data from numerous clinical trials indicate that monotherapy will not provide long-term efficacy in aggressive cancers [39]. Therefore, combination therapies that target multiple pathways or networks will be required to improve survival outcomes in aggressive cancers such as OS.

To improve the clinical outcome in aggressive OS, it will be necessary to identify combination therapies that are well-tolerated and target a network of nodes or hub genes in perturbed signaling pathways. To this end, a systems biology approach that incorporates “OMICs” data as the foundation for an evaluation of complex biological networks via interdisciplinary collaborations (clinical, pre-clinical, and bioinformatics) can be employed to develop hypothesis-driven approaches for experimental design and analysis [40,41]. “OMICS” refers to a full comprehensive study of a biological sample that ranges from its genome, epigenome, transcriptome proteome, metabolome, and its microbiome [42]. This approach helps understand the etiology of the disease and different layers of the “OMICs” approach can be investigated depending on the question of interest [42]. Using the “OMICS” approach in conjunction with “Systems biology” method will help understand the molecular complexity of OS and help identify possible therapeutic networks. Systems biology approach involves large-scale data mining, network analysis, and validation via preclinical in-vivo modeling, all of which can help advance the rational use of combination therapy that is efficacious, but also safe [43]. In the present study, our overall objective was to integrate CNVs and corresponding gene expression data with overall survival in OS to identify prognostic risk signatures and to functionally validate a therapeutically actionable risk signature. Through our integrative bioinformatic analysis, we identified a large number of CNVs that correlate with poor overall survival across the genome. To prioritize and provide rationale for what to investigate in preclinical models of pediatric and AYA OS, we used a “tiered funnel” approach. Four criteria were utilized to select the candidate CNV with a poor prognostic signature that would be investigated in more detail: (1) Correlation of the CNV with poor overall survival in a majority of the OS patients; (2) association of the genes expressed from the CNV with a dysregulated network that can be dually targeted; (3) availability of clinically relevant inhibitors targeting the dysregulated network; and (4) identification of molecularly characterized models of OS harboring the pertinent CNV signature.

The MYC-RAD21+ signature met all four criteria and was prioritized for a preclinical investigation. This signature significantly correlated with increased gene expression and poor survival in >90% of OS patients in the Therapeutically Applicable Research to Generate Effective Treatments (TARGET) database [44]. Both of these multifunctional proteins have biological relevance in a number of signaling networks that are dysregulated in cancer [45,46,47,48,49]. Notably, MYC and RAD21 overexpression can serve as a biological indicator of replication stress (RS) that is therapeutically actionable [50]. While high levels of RS can prevent cancer progression by inducing cell death [3,51], moderate RS may allow for an accumulation of CNVs that elicit gene dysregulation and, ultimately, contribute to tumorigenesis [51]. Therefore, modulation of this dysregulated biological process is very attractive for anti-cancer therapy, especially for genetically complex cancers such as pediatric sarcomas [51,52].

MYC amplification is one of the most common prognostic biomarkers for patients with OS and serves as an attractive therapeutic target [53]. While MYC is a well-known oncogenic driver of OS growth and metastasis, it also plays a critical role in RS by regulating cell cycle machinery, as well as co-localizing to specific nuclear sites associated with early origins of replication [47,52,54]. Notably, MYC has been challenging to directly target. However, inhibitors have been developed for bromodomain and extra-terminal domain (BET) proteins (BRD2,3,4) that block the binding of BETs to acetylated lysines on nucleosomes and decrease the gene expression of transcription factors such as BCL6 and MYC [55,56]. While in some molecular contexts, MYC expression can be indirectly inhibited by bromodomain and extra-terminal domain inhibitors (BETi) [57], this is not always the case, as demonstrated in glioblastoma [58]. Regardless, the use of BETi may be promising for treating pediatric OS because an MYC-independent BET function can also affect the expression of RS-related targets [59,60]. Our rationale for selecting MYC-RAD21+ as the molecular signature to target was further supported by the fact that a network link between MYC and RAD21 exists, for RAD21 gene expression can be regulated by MYC [61]. RAD21 is often overexpressed in highly replicative cancers [61] and it has been reported that RAD21 is amplified in OS and other cancers [62,63]. RAD21 functions as a chromatid cohesion subunit that plays an essential role in DNA replication and repair. Moreover, it has been reported that MYC activation in RAD21 silenced cells results in increased DNA damage and replicative stress [47,61]. Although RAD21 cannot be targeted therapeutically, there is emerging evidence that an increased RAD21 protein level correlates with an increased sensitivity to CHK1 inhibitors (CHK1i) [64]. Additionally, mechanistic links may exist between RAD21 and CHK1, for example, in yeast, CHK1 can directly phosphorylate RAD21 [65]. Furthermore, MYC can also directly increase CHK1 gene expression in some cancers [65]. Further rationale for prioritization of the MYC-RAD21+ signature in OS is provided by Rohban et al. They demonstrate that RAD21 may restrain MYC-induced replication stress by cohesion-mediated DNA synthesis to enable faithful clonal expansion of tumor cells [61]. This may help explain why MYC and RAD21 are frequently co-amplified in many cancers [61]. Another reason for MYC and RAD21 co-amplification/co-copy number gain could also be that MYC and RAD21 are in close proximity on the 8q24 amplicon [45].

In this study, we employed a systems biology approach that identified a large number of prognostic signatures that correlate with poor overall survival. As outlined above, the MYC-RAD21+ signature, due to its functional association with RS, was prioritized for the study in preclinical models of pediatric and AYA OS. We tested the hypothesis that in OS models harboring the MYC-RAD21+ risk signature, the dual inhibition of RS regulators BET (OTX-015) + CHK1 (SRA737 or LY2606368) would be efficacious and well-tolerated. Preclinical and clinical (NCT03205176) evaluation of BETi/OTX-015 is under evaluation in refractory solid tumors such as OS [66]. In addition, CHK1i/SRA737 is also being employed in a clinical trial for advanced cancers (NCT02797964, [63]), and CHK1i/LY2606368 is under evaluation by Cincinnati Children’s Hospital Medical Center in a Phase 1 Study in mutant TP53 refractory solid and liquid tumors [8,62,67]. To the best of our knowledge, our study is the first to integrate the analysis of CNVs, gene expression, and OS patient survival data to prioritize actionable therapeutic targets and functionally validate drug responses in preclinical models of OS with the MYC-RAD21+ signature.

## 2. Results

### 2.1. Significant Correlation between Gene Expression Profiles and Survival in Two Independent OS Datasets

A Cox proportional hazards model illustrated the correlations between gene expression and overall survival from two independent datasets. TARGET (*n* = 85 OS patients; Appendix A) [44] and Gene Expression Omnibus (GEO) GSE16091 database (*n* = 34 OS patients; Appendix A) [68] datasets were utilized to evaluate correlations between gene expression and overall survival. This query was conducted on 16,178 genes available in TARGET and 10,704 genes in the U133A platform within the GSE16091 database. Among the 16,178 total genes evaluated in TARGET, the expression of 1420 genes showed a significant correlation with OS overall survival (Figure 1a, *p* < 0.05; Appendix A). In the GEO GSE16091 database, the expression of 590 genes out of the 10,704 total genes significantly correlated with overall survival (Figure 1a, *p* < 0.05; Appendix A). Collectively from the two datasets, 71 expressed genes were found in both databases that showed consistent correlations with OS overall survival, with a false discovery rate of 17% based on their *p* values and their direction of association (i.e., positive/negative association between gene expression and overall survival) (Figure 1a, see Venn diagram, Appendix A and Appendix A—see genes in red text). Overall survival in OS patients was analyzed based on demographic and clinical variables in TARGET (Figure 1b). In contrast to gender, age, or race, a diagnosis of metastatic disease was the only variable that was significantly associated with poor overall patient survival. Patients with non-metastatic OS at diagnosis had a lower mortality risk than patients diagnosed with metastatic disease at diagnosis (Figure 1b; HR = 0.21, *p* < 0.0001). No demographic and clinical variables were available for analysis in the GSE16091 OS dataset.

### 2.2. CNVs Significantly Correlated with the Overall Survival in OS Patients

The TARGET database not only includes gene expression and overall survival data, but also encompasses CNV data, which is absent from the GEO GSE16091 dataset. Among the overlapping genes identified in both the TARGET and GSE16091 datasets, a significant correlation was observed between the expression of 40 of the 71 overlapping genes with corresponding CNVs in TARGET (*p* < 0.05; Appendix A—see genes in purple bold text). Within these 40 genes, 14 genes not only showed a significant correlation between gene expression and CNVs, as mentioned above, but also a significant correlation between gene expression and overall survival, as well as between CNVs and overall survival (*p* < 0.05 with a false discovery rate of 8.2%; Figure 2 and Appendix A—see genes in green bold text). CNVs such as the presence of copy number gains were observed in *MYC*, *RNF139*, *SQLE*, *E1F3H*, *UBRR5*, and *IGF1R*, whereas *MLF2*, *TXK*, *PHB*, *LZTS1*, *DDX21*, *FBXW4*, *CHML,* and *MAPK8IP3* showed more copy number deletions (Figure 2). Among the 14 genes, 10 genes (*MYC*,*RNF139*, *SQLE*, *E1F3H*, *UBR5*, *IGF1R*, *LZTS1*, *DDX21*, *FBXW4*, and *CHML*) have HR values > 1, which signifies that those particular genes have a copy number gain and increased gene expression that increase the risk for poor overall survival (Figure 2). The four remaining genes (*MLF2*, *TXK*, *PHB*, and *MAPK8IP3*) had an HR value < 1, which is indicative of a copy number deletion and decreased gene expression that result in an increased risk for poor overall survival (Figure 2). Moreover, the expression of the non-receptor tyrosine kinase TXK gene, which plays a role in regulating T-Helper 1 (Th1) cells [69], was decreased by 50-fold at the transcriptome level in OS samples compared to normal bone tissue samples (Figure 2). We further compared the expression of the 14 genes mentioned above in OS samples and non-neoplastic primary normal osteoblasts from an independent GSE14359 dataset (Figure 2). In particular, there was a 310-fold increase in RNA expression of the tumor suppressor gene on chromosome 8p—LZTS1. LZTS1, also known as FEZ1, is a tumor suppressor gene that has been reported to be at decreased levels in various kinds of tumors compared to normal samples [70,71]. LZTS1 produces multiple transcripts, resulting in various isoforms with assorted functions that still need to be investigated [72]. Moreover, while we could observe increased mRNA levels compared to OS samples, the LZTS1 protein levels are not known. The nine other genes were increased by 1.2~15.0 fold [70] (Figure 2). MYC exhibited four-fold change and displayed the most significant correlation between CNV and overall survival, as well as gene expression and overall survival. The role of the other 13 genes which harbor CNVs in OS requires further investigation (Figure 2) [21,67].

### 2.3. Mapping of CNVs to Their Corresponding Genes that are Associated with Overall Survival in OS Patients

The remaining bioinformatic analysis focused on the TARGET database, as it includes CNV, gene expression (GE), and clinical outcomes data. Using the TARGET dataset, CNVs that were identified in TARGET were mapped to 26,783 genes (Hg19 genome, https://www.genenames.org/) to standardize the gene annotations and verify the gene nomenclature. Their correlations with the overall survival were analyzed using the Cox proportional hazard model. The chromosome distributions of various genes whose CNVs are significantly associated with overall survival are depicted in Table 1, *p* < 0.05. Chromosomes 8, 10, 11, 15, 16, and 17 have significantly more genes with CNVs that are associated with patient overall survival, and their chromosome enrichment *p*-values are 1.05 × 10^−72^, 1.44 × 10^−163^, 3.13 × 10^−166^, 1.33 × 10^−28^, 4.18 × 10^−70^, and 4.02 × 10^−7^, respectively (red in Table 1). In chromosome 8, the overall survival-associated genes consisted of a higher number of copy number gains than deletions (188 vs. 160, Table 1) compared to other chromosomes. In the other five CNV-enriched chromosomes (10, 11, 15, 16, and 17), there were more deletions than copy number gains.

### 2.4. Multiple CNVs Associated with Increased Gene Expression and Poor Overall Survival in OS Are Present on Chromosome 8

Cox regression analysis conducted on OS patients from the TARGET database (*n* = 85) suggested that CNVs in 2640 genes significantly correlated with an increased risk for a poor prognosis in OS patients. CNVs in 260 genes out of the 2640 were present on chromosome 8 (Figure 3; Appendix A). Furthermore, the top 10 genes with CNVs that were significantly associated with an increased risk for relapse in OS patients were all present on chromosome 8 (Figure 3; Appendix A). The MYC-RAD21+ signature was associated with upregulated gene expression and poor overall survival for OS patients based on OS patient data in the TARGET database (Figure 3). Kaplan Meier survival plots generated using the TARGET database demonstrated that MYC copy number gain was significantly associated with decreased overall survival among OS patients (log-rank test *p* = 0.04, HR = 2.05) (Appendix A). Kaplan Meier Survival Plot for RAD21 using log-rank test (p = 0.011, HR = 2.70) indicates that copy number gain or normal copy number is associated with decreased overall survival in OS patients 1 (log-rank test *p* = 0.011, HR = 2.7) (Appendix A).

### 2.5. Ingenuity Pathway Analysis of Genes with CNVs Significantly Associated with Overall Survival in OS Patients

A network analysis was performed on the genes of 2640 CNVs that significantly correlated with poor overall survival in TARGET using Ingenuity Pathway Analysis (IPA) (Appendix A, CNV TARGET list). As expected, MYC had a high degree of connectivity with a large number of predicted protein-protein interactions (Appendix A, gene degree by STRING database). Other top interacting network proteins included HRAS, EGFR, and MAPK (Appendix A). RAD21 exhibited a moderate degree of connectivity. Many of the CNVs were related to genes associated with connective tissue disorders (Appendix A). Among the top disease subnetworks designated by IPA, MYC CNVs were associated with skeletal/muscular disorders and RAD21 CNVs were associated with connective tissue disorders (Appendix A). Pathway enrichment analysis of CNVs from TARGET that correlated with poor overall survival indicated a number of links between MYC and MAPK signaling. RAD21 was associated with mitotic and cell cycle pathways (Appendix A).

### 2.6. Prioritization of the MYC-RAD21+ Risk Signature for Investigations in Pre-Clinical Models of Pediatric and AYA OS

The -OMICS analyses we interrogated were genome wide and provided a number of potential targets for investigation. These analyses not only confirmed previous data obtained from others, but also integrated independent data sets to enhance the identification and prioritization of CNVs that significantly correlate with GE and overall survival in pediatric and AYA OS patients. As mentioned above, there were 260 genes with CNVs on chromosome 8 that correlated with increased GE and poor survival in OS using the TARGET database. The MYC-RAD21+ signature on chromosome 8q was among the top 10 most significant copy number gains that correlated with increased GE and poor OS survival in >90% of the OS patients. Increased levels of MYC and RAD21 can serve as indicators of dysregulated RS in cancer [50]. Therefore, we hypothesized that by inhibiting key regulators of RS that control its severity, RS would increase to intolerable levels and lead to OS cell death. Targeting RS in OS has not been studied in great detail and a number of promising small molecule inhibitors to RS regulators are in clinical trials [51,54,58,60,63]. In these studies, we utilized in-vitro and in-vivo models of pediatric and AYA OS that harbor the MYC-RAD21 CNV+ risk signature to explore the efficacy of inhibiting two RS regulators: BETs and CHK1.

### 2.7. In Vitro Assessment of Cell Growth in MYC-RAD21+ Pediatric OS Lines Following the Pharmacological Inhibition of BET Proteins and CHK1

A panel of established pediatric OS cell lines (Saos2, Saos2-LM7, MG63, G292, and U2OS), as well as an OS xenoline derived from the TT2-77 patient-derived xenograft (PDX), were selected for in vitro screening with BETi and CHK1i. Whole genome sequencing (WGS) of OS cell lines indicated that copy number gains of MYC and RAD21 ranged from 3 to 12 and 0 to 7 copies, respectively (Appendix A). Copy number gains do not necessarily correlate with increased gene and/or protein expression due to transcriptional and post-transcriptional regulation [73]. Therefore, the expression of pertinent targets (MYC, RAD21, CHK1, and BET proteins) was confirmed in all OS cell lines by Western blot (Appendix A and Appendix A). In the U2OS cell line, while an RAD21 copy number gain was not observed (Appendix A), RAD21 protein expression was still present at higher levels compared to normal human osteoblasts (NHOSTs) (Appendix A). In the TT2 xenoline, proteins of interest were also compared to the original PDX from which it was established (TT2-77). In contrast to the PDX tissue, the proteins of interest in the TT2-77 xenoline were expressed at higher levels. This is likely attributable to murine stromal components estimated from sequencing data to be ~10%. In addition, the TT2 line is presumably a subpopulation of the human tumor cells in the PDX. Notably, BRD4 was lower in the PDX, which is likely due to the fact that the BRD4 antibody only recognizes human BRD4. The other antibodies used for Western blot are specific for human and murine proteins. The BETi/OTX-015 and CHK1i/LY2606368, as single agents, potently blocked cell growth in all of the OS cell lines and the TT2-77 xenoline at clinically relevant concentrations (Figure 4). However, Saos2 and Saos2-LM7 cells were relatively more resistant to CHK1i/SRA737 than the other OS cell lines (Figure 4). The levels of BETi/OTX-015 clinically achievable in human plasma are dose regimen-dependent, with plasma exposure of up to 1–2 µM OTX-015 [58]. The C_max_ of CHK1i/SRA737 that can be safely obtained in the human plasma was approximately 6 µM CHK1i/SRA737 [63]. In a phase I study (NCT02203513), an average plasma concentration of 0.1–0.2 µM CHK1i/LY2606368 could be obtained over a 24 h time interval [74].

To determine to what extent the simultaneous inhibition of BET proteins and CHK1 blocks growth, MYC-RAD21+ established OS cells and the TT2-77 xenoline were exposed to BETi/OTX-015 plus CHK1i/SRA737 (Table 2a) or BETi/OTX-015 plus CHK1i/LY2606368 (Table 2b) over a broad range of dose ratios. By evaluating a range of dose combinations, insights can be obtained into inhibitor concentrations that need to be achieved in vivo to efficiently inhibit cell growth. Combination drug effects were analyzed for potency and efficacy synergy. An analysis of potency synergy by the combination index (CI) indicated a high degree of synergy in Saos2 and Saos2-LM-7 cell lines (CI < 1.0 at 20–80% growth inhibition), while antagonism was observed in the G292, MG63, U2OS, and TT2-77 cell lines (CI > 1.0 at 20–80% growth inhibition) (data not shown). An analysis of efficacy synergy by Bliss independence indicated that all OS cell lines exhibited at least additive cell-growth inhibition at specific dose ratios of the inhibitors. An additive to the synergistic inhibition of growth in Saos2 or Saos2-LM7 following exposure to BETi/OTX-015 with CHK1i/SRA737 (Table 2a) or CHK1i/LY2606368 (Table 2b) was observed at clinically achievable concentrations over a broad range of dose ratios. In other OS cell lines (G292, MG63, U2OS, and TT2-77 xenoline) exposed to BETi/OTX-015 with CHK1i/SRA737 (Table 2a) or CHK1i/LY2606368 (Table 2b), certain dose ratios resulted in additive growth inhibition at clinically achievable concentrations, while others conferred antagonism, presumably because these OS lines were already quite sensitive to single-agent therapy at clinically achievable concentrations of the inhibitors (see IC50 values, Figure 4).

### 2.8. In Vitro Dual-Inhibition of BET and CHK1 Signifcantly Induces Apoptosis in Saos2 Cells

As shown above, a blockade of the BET function decreased the resistance to CHK1 inhibition, resulting in a high degree of synergistic growth inhibition at clinically achievable doses in Saos2 and Saos2-LM7 cells (Table 2a,b). We next determined whether synergistic cell-growth inhibition correlated with apoptotic cell death. Saos2 cells were treated with a vehicle, single agent, or BETi/OTX-015 (0.31 μM) + CHK1i/SRA737 (3.75 μM), which are clinically-achievable doses that induce synergistic cell-growth inhibition (Table 2a). BETi/OTX-015 + CHK1i/SRA737 combination treatment exhibited a significant increase in activated Caspase-3/7 compared to vehicle and single-agent treatments following 2 days of exposure (Figure 5a) that correlated with increased PARP cleavage (a downstream apoptosis marker) [75] (Figure 5b; Appendix A). In addition, flow cytometric analysis of apoptosis by Annexin V and Propidium Iodide staining indicated that by 4 days post-treatment, there was a significant increase in early apoptotic (Figure 5c) and late apoptotic/necrotic (Figure 5d) Saos2 cells treated with BETi/OTX-015 + CHK1i/SRA737 versus the vehicle or single agent. By 5 days post-treatment, only minimal numbers of viable Saos2 cells remained on plates treated with the combination BETi/OTX-015 + CHK1i/SRA737 compared to the vehicle or single agent (Appendix A).

### 2.9. BET Inhibition does not Decrease MYC Protein Expression but does Inhibit OS Cell-Growth by BRD4-Dependent Mechanisms

There are likely a number of underlying mechanisms that contribute to BETi/OTX-015 + CHK1i/SRA737-mediated growth inhibition and increase apoptosis. As previously mentioned, MYC has been challenging to therapeutically target. Inhibitors of BET proteins, such as BETi/OTX-015, bind to the acetyl-binding grooves of BET proteins (BRD2, 3, 4) and release BET proteins from the chromatin, resulting in chromatin compaction and the decreased transcription of genes such as *MYC* [76,77,78]. It has also been reported that the inhibition of BET-BRD4 results in MYC-dependent growth inhibition and apoptosis in some OS cell lines [8]. Due to this and the fact that MYC can directly increase CHK1 gene expression in some cancers [65], we hypothesized that BETi/OTX-015 would down-regulate MYC and contribute to decreased CHK1 protein levels. To test this hypothesis, Saos2 cells were treated with a vehicle, single agents, or a combination of BETi/0TX-015 and CHK1i/SRA737. However, there was no difference in MYC or CHK1 levels at 4 or 16 h post-exposure for single or combination treatment. In addition, the levels of BET proteins (BRD2-4), RAD21, and the loading control (vinculin) were unchanged following drug exposure (Figure 6; Appendix A). Activation of the DNA damage response was evident for exposure to single-agent CHK1i/SRA737 and combination BETi/OTX-015 + CHK1i/SRA737 increased the AKT-mediated phosphorylation of CHK1 at serine 345 *p*-CHK1 S345 (a pharmacodynamic biomarker of DNA damage [79]) and γH2AX (a pharmacodynamic biomarker of replication stress and double-strand DNA breaks [80]). These results suggest that BETi/0TX-015-mediated effects on MYC protein levels do not represent a mechanism of action that contributes to synergistic cell-growth inhibition (Table 2) and apoptosis (Figure 5) induced by combination BETi + CHK1i in Saos2 OS cells.

We next focused on the role of BRD4 in regulating target gene expression, as well as its role in specifically increasing the sensitivity to CHK1 inhibition. BRD4 transient knock-down in Saos2 cells by siRNA was validated by Western blot (Appendix A). The knockdown (KD) of BRD4 did not have an impact on the levels of MYC protein, confirming that BRD4 does not regulate MYC protein levels in the molecular context of Saos2 cells (Appendix A). In addition, BRD4 KD did not affect the protein expression of BRD2, BRD3, or CHK1 (Appendix A). BRD4 was silenced in Saos2 cells and subsequently treated with CHK1i/SRA737 and cell-growth evaluated (Figure 7). Silencing BRD4 in Saos2 cells resulted in decreased cell growth. In the presence of CHK1i/SRA737, cell growth was further decreased, suggesting that BETi/OTX-015 + CHK1i/SRA737-mediated growth inhibition in Saos2 cells is partially dependent on BRD4.

### 2.10. Dual-Inhibition of BET and CHK1 in a Saos2 Cell Line-Derived Xenograft (CDX) Model Increases the Probability of Survival over Time

To determine if combination BETi and CHK1i is efficacious and safe in vivo, we first used the Saos2 CDX model. This model was selected for an in-vivo study since the in-vitro screening data predicted that the inhibition of BET and CHK1 can synergistically block Saos2 cell growth. SRA737 was prioritized as the CHK1i for in vivo studies due to its safety profile [63], ease of formulation, and oral delivery. In Figure 8a, the tumor volumes are presented when all mice in each group are still under study and have not reached the endpoint tumor volume. During the dosing period, tumors continued to grow in all treatment groups (Figure 8a). All the mice in the vehicle and single-agent groups reached their endpoint tumor volume of 1500 mm^3^ during the dosing period (Figure 8b). In contrast, none of the mice in the BETi/OTX-015 + CHK1i/SRA737 combination therapy group reached the endpoint tumor volume during the dosing period (Figure 8b; Appendix A). The Kaplan–Meier survival plot indicated that the BETi/OTX-015 + CHK1i/SRA737 combination therapy resulted in a statistically significant increase in the probability of survival compared to vehicle or single-agent therapy (Figure 8b; Appendix A). The median survival was 41 days for the vehicle group, 34 days for the CHK1i/SRA737 treatment group, 48 days for the BETi/OTX-015 group, and 65 days for the combination treatment group (Figure 8b). The single and combination treatments were also well-tolerated, as assessed by body weight (Appendix A).

### 2.11. Dual-Inhibition of BET and CHK1 in a PDX Model of Relapsed OS Signficantly Arrested Tumor Growth During the Dosing Period and Increased the Probability of Survival over Time

We next compared the effect of single-agent BETi or CHK1i versus combination BETi + CHK1i on the growth of a relapsed OS TT2-77 PDX. Whole genome sequencing indicated that both the OS biopsy at initial diagnosis and the TT2-77 PDX generated from a tumor specimen obtained at resection three years later harbor the MYC-RAD21+ signature (original biopsy and TT2-77 PDX = 4 copies per amplicon, Appendix A). As shown previously in Figure 4, the growth of the TT2-77 xenoline was significantly inhibited at clinically relevant concentrations of single-agent BET and CHK1 inhibitors. Bliss analysis indicated an additive effect on growth inhibition at select dose ratios of combination BET and CHK1 inhibitors (Table 1). The in-vivo efficacy data revealed a more pronounced effect on growth inhibition. In TT2-77 PDX mice treated with single-agent BETi or CHK1i, the tumor growth kinetics slowed compared to vehicle-treated mice, but still continued to grow during the dosing period (Day 15–Day 42 post-implant). In mice treated with BETi/OTX-015 + CHK1i/SRA737 combination therapy, tumor growth was significantly halted compared to vehicle or single-agent therapy (Figure 9a). BETi/OTX-015 + CHK1i/SRA737 combination therapy resulted in a statistically significant treatment-related decrease in the TT2-77 tumor volume during the dosing period (days 15–45) (Figure 9a). Once therapy was discontinued, tumor volumes began to increase in the mice treated with BETi/OTX-015 + CHK1i/SRA737 (Appendix A). Kaplan–Meier survival analysis demonstrated a significant increase in the probability of survival in mice treated with BETi/OTX-015 + CHK1i/SRA737 (Figure 9b; Appendix A). The median survival was 55 days for the vehicle group, 71 days for the CHK1i/SRA737 treatment group, 64 days for the BETi/OTX-015 group, and 92 days for the combination treatment group (Figure 9b). The single and combination treatment was also well-tolerated, as assessed by body weight (Appendix A), bone marrow cellularity (Appendix A), and an analysis of vital organ integrity via H&E staining and pathology analysis (Appendix A).

## 3. Discussion

Despite advancements in multimodal therapeutic approaches, OS still remains challenging to treat due to its genetic heterogeneity and histological variability [3]. Additionally, treatments deemed suitable for adult OS patients may not always be appropriate for pediatric and AYA patients with OS due to differences in tumor biology, genetic alterations, and drug pharmacokinetics in adults versus children [3,81]. The rarity of OS makes it especially challenging to accelerate the development of new drugs to treat this disease due to the cost of drug development and the small number of pediatric patients available to enroll in clinical trials [82]. Therefore, new approaches are needed that utilize population data to identify relevant biomarkers, risk signatures, and targetable pathways that may help guide therapeutic selection for improving the overall survival of pediatric and AYA OS patients.

OS, which is the most common pediatric bone cancer, is characterized by heterogeneous chromosomal instabilities (CINs) ranging from aneuploidies (numerical–CIN) to chromosomal gains or losses (structural- CINs) that can give rise to CNVs [17,83,84,85]. Since CNVs can encompass a large number of genes, identifying key driver genes is difficult. Others have reported that >15% heritable variation in gene expression is due to CNVs [86]. Some of the genes identified by CNV analyses may serve as potential therapeutic targets or biomarkers of biological processes that can be targeted. However, there is limited knowledge on how CNVs are associated with gene expression and patient survival. It is also not clear if poor prognostic CNV signatures linked to increased gene expression can predict therapeutic responses to therapy. Therefore, integrating CNVs with gene expression and patient survival can help identify and prioritize genetic alterations that may have a functional oncogenic or driver effect, and are actionable [86]. One of the key driving forces that promotes the formation of CNVs is RS, which is a process involving the stalling or collapse of replication forks to stop DNA replication [87,88,89,90]. RS is an important hallmark in tumorigenesis and is a process unique to cancer cells, thereby making the modulation of this biological process very attractive for anti-cancer therapy [51,52]. In normal healthy cells, if DNA is damaged, then cells either repair it or induce cell death [51]. However, in cancer cells where cell cycle checkpoints or DNA repair pathways are defective, cells attempt to replicate the damaged DNA, which results in fork stalling and the activation of downstream RS pathways [51]. Several extrinsic factors (ultraviolet radiation or genotoxic agents), as well as intrinsic factors (increased oncogene activation), initiate RS [51]. While mild levels of RS induce CIN and contribute to tumorigenesis, high levels of RS may actually protect against cancer progression by inducing cell death [51]. Therefore, identifying actionable biomarkers within the RS pathway that contribute to the complex genetic landscape of pediatric sarcomas and increasing RS to eradicate cancer cells may be an effective therapeutic strategy [3].

The objective of the present study was to expand upon our previous work with a specific focus on pediatric OS, in order to identify, prioritize, and functionally validate molecular biomarkers of therapeutic responses using rationally designed combination therapies [11]. In the current study, we utilized a systems biology approach to interrogate various tiers of pathogenesis (CNVs, gene expression, and patient outcomes) in our OS system. As depicted in Figure 10, we used datasets from TARGET and GSE16091 to compare and correlate CNVs with GE. Additionally, GSE14359 dataset was utilized to assess GE in OS patients versus normal non-neoplastic primary normal osteoblasts from an independent GSE14359 dataset (Figure 2, Figure 10). Results from all 3 datasets contributed to the network pathway analysis reported in this study. The global genome was interrogated to identify CNVs that correlate with gene expression and overall survival. We then used specific criteria to narrow our focus to a statistically significant and actionable molecular signature that can be targeted in pediatric and AYA OS. Therefore, the evaluation and integration of these multi-scale biological networks resulted in prioritization of the MYC-RAD21+ for therapeutic intervention in experimental OS models [91].

Our results from the functional annotation analysis of CNV-associated differentially expressed genes and protein-protein interaction-based pathway analysis indicate that MYC and RAD21 are significant hub proteins. As mentioned above, the MYC-RAD21+ signature not only correlates with poor overall survival in OS, but is also an indicator of RS, which is a biological process that can promote genome instability and likely contributes to the development of CNVs in OS [50,59]. There are a number of mechanistic links between MYC, RAD21, BETs, and CHK1. MYC and RAD21 play a role in RS and the DNA damage response (DDR) pathways [92,93]. MYC, a transcription factor involved in the pathogenesis of OS, has been known to be overexpressed and amplified in localized and metastatic OS, and contributes to therapeutic resistance of the disease [47,94]. Notably, Shi et al. used biological and functional enrichment analysis to demonstrate that MYC is part of the risk signature for OS that is associated with metastatic disease and may help predict an increased chance of relapse [95]. Our study not only delves into identifying poor prognostic signatures that are therapeutically actionable, but also functionally validates the MYC-RAD21+ signature as a biomarker of the response to RS-targeted therapy using in-vitro and in-vivo MYC-RAD21+ pediatric OS models.

In addition, MYC-independent effects of BETs can also affect the expression of RS-related targets [59,60]. For example, BET-BRD4 can directly contribute to RS by associating with pre-replication factors to induce genes involved in DNA repair [59,60]. It is possible that due to the close proximity of MYC and RAD21 on chromosome 8, RAD21 may simply serve as a companion biomarker with no major biological relevance. However, recent studies have demonstrated the functional significance of RAD21 in OS by highlighting that OS cell apoptosis involves the cleavage of RAD21 [46]. As previously stated, CHK1 contributes to the DDR, and in melanomas, increased RAD21 expression associated with an increased sensitivity to CHK1i [64]. Notably, CHK1 has been reported to phosphorylate RAD21, as well in other model systems [65]. Furthermore, RAD21 can also regulate MYC-induced RS [61].

In the present study, we investigated the effectiveness of targeting RS in MYC-RAD21+ pediatric OS cell lines and OS xenograft models. As single agents, BETi/OTX-15, CHKi/SRA737, and CHK1i/LY2606368 were quite potent in terms of cell–growth inhibition in most OS cell lines. The efficacy of the combination BETi + CHK1i, as analyzed by Bliss Independence, revealed differences in the extent of growth inhibition in a panel of OS cell lines and the TT2-77 xenoline. Among these cell lines, Saos2 and Saos2-LM7 were the most resistant to CHK1i/SRA737 single-agent therapy and BETi increased the sensitivity to CHK1i/SRA737, resulting in synergistic cell growth inhibition at clinically achievable concentrations. Dual-inhibition of BET and CHK1 resulted in significant apoptosis that was dependent on BRD4, but did not affect MYC protein levels. In vitro studies on ovarian cancer cells have also shown that therapy involving BET + CHK1 inhibition may be a promising treatment modality [96]. This study demonstrated that inhibition of BRD4 via siRNA or the BETi JQ1 increased heterochromatin protein 1 (HP1) in ovarian cancer cells which limited the DDR and increased sensitivity to CHK1i [96]. Studies are in progress to gain a detailed understanding of the underlying mechanisms of action contributing to the significant growth inhibition observed in pediatric and AYA OS models following exposure to BET + CHK1 inhibition.

In this study, we not only integrated the MYC-RAD21+ signature with CNVs, gene expression, and patient survival data, but we also linked these data with preclinical efficacy and safety data using in vivo models of OS. In a pediatric OS CDX model, as well as a PDX model of relapsed AYA OS, we demonstrated that the simultaneous targeting of BET and CHK1 increased the probability of survival and was well-tolerated. As mentioned previously, the TT2-77 PDX derived from a patient with relapsed OS harbors the MYC-RAD21+ signature. Of critical importance, the original diagnostic biopsy sample from this patient also harbored the MYC-RAD21+ signature, which emphasizes the importance of detecting poor prognostic signatures at diagnosis, which could ultimately lead to innovative clinical trial approaches in OS patients that target poor prognostic signatures up front and prevent relapse in the first place. Studies are in progress to optimize BETi + CHK1i combination therapy in the context of standard-of-care therapy, understand BRD4-dependent mechanisms of action, and study the tumor adaptive response over time. Moreover, studies are in progress to evaluate combination therapies targeting multiple tiers of RS pathways, such as dual-inhibition of BET with ATR or WEE1 [47,48,88]. It will also be informative to evaluate whether BET proteins BRD2 and BRD3 play a role in blocking cell growth. [47,88,97]. It is not clear at this time whether the MYC-RAD21 signature is the sole predictor of the therapeutic response to BETi + CHK1i. It is likely that the cellular context and not just the MYC-RAD21 signature will dictate the response to BETi + CHK1i. Moreover, MYC, BETs, RAD21, and CHK1 protein levels could dictate the sensitivity to the combination of BETi + CHK1i, independent of the MYC-RAD21 CNV status and this is under investigation. In addition, testing novel strategies identified in this study that target other signatures of poor prognosis in pediatric OS, such as the MAPK signaling network, are ongoing. This study provides new information on the safety and efficacy of concomitantly inhibiting BET and CHK1 with the goal of, ultimately, improving the prognosis in children and AYA with aggressive OS, the majority of whom now die of this devastating disease.

## 4. Materials and Methods

An overview of the systems biology data analysis for osteosarcoma data is presented in the following figure. It started from three osteosarcoma data sources, including TARGET, GSE16091, and GSE14359. Then, correlative analyses were performed for copy number variation (CNV), gene expression (GE), and overall survival (OS), and significant genes and prognostic biomarkers, were selected and overlapped among different correlative analyses. The pathway analysis was performed to explore the underlying molecular mechanisms among these prognostic biomarkers.

### 4.1. Osteosarcoma Genomics Datasets for a Systems Biology Approach

One hundred and three OS were collected from the following publicly accessible databases: NCBI-Gene Expression Omnibus (GEO) (https://www.ncbi.nlm.nih.gov/sites/GDSbrowser/) serials ID GSE16091 and Therapeutically Applicable Research to Generate Effective Treatments (TARGET) (https://ocg.cancer.gov/programs/target). Two osteosarcoma datasets include CNV, mRNAs, and associated clinical patients’ demography.

TARGET: A total of 89 gene expression profiles of OS were obtained from TARGET [44], which was tested by Affymetrix Human Exon 1.0 with 16,178 genes and 88 chromosome copy number analyses and loss of heterozygosity profiles, which was tested by Affymetrix SNP 6.0 Array. Of these gene expression profiles, 85 patients had complete datasets and were included in the analysis. GSE16091: In total, 34 gene expression profiles of OS were obtained from GSE16091 [68], which was tested by Affymetrix Human Genome U133A with 10,704 genes. GSE14359: In order to characterize OS disease genes, we collected GEO series ID GSE14359 [98]. This included 10 expression data from five frozen conventional OS patients, involving 10 conventional OS tissues and two non-neoplastic primary normal osteoblasts cells, using Affymetrix HG U133A microarrays.

### 4.2. Gene Annotations for a Systems Biology Approach

The HUGO Gene Nomenclature Committee (HGNC, http://www.genenames.org/) database provides researchers with standard gene names for the human genome to avoid the complexity of multiple overlapping and conflicting nomenclature systems. The database currently consists of around 24,000 genes and their corresponding approved gene symbols. Each gene has a unique HGNC ID, which makes it easier to identify the gene type. Genes were also annotated with other information, including gene synonyms, uniprot IDs, refseq IDs, previous gene symbols, and a functional description about each gene, all of which aid in integrating information from the NCBI or other databases [99].

### 4.3. Correlative Analysis between Gene Expression and Patient Overall Survival

Gene expression high-throughput data preprocessing: Microarray Human Genome U133A and Affymetrix Human Exon 1.0 data (.cel files) was processed and normalized using the Robust Multi-array Average (RMA) [100] algorithm. The normalized log2 ratio (healthy/tumor) on probe-sets was annotated for genes for further analysis. The data were analyzed in R [101].

Cox regression (or proportional hazards regression) [102] was analyzed using the Coxph() in R. This analysis investigates the effect between the gene expression or CNVs and patient survival. The confounding factors of race and metastasis status were justified in the multivariate survival model [103].

### 4.4. Correlative Analysis between CNV and Patient Overall Survival

CNV analysis was conducted using Partek Genomics Suite (Partek, Inc., St. Louis, MO, USA). The Genomic Segmentation method from Partek Genomics was used to detect break points and obtain CNV calls from the log intensities of the Affymetrix Genome-Wide Human SNP Array 6.0. Default parameters were used for the detection of copy number gain and deletions: Minimum genomic markers = 10; *p*-value threshold = 0.001; signal to noise = 0.3; and diploid copy number = 1.72 to 2.78. In other words, CNV values smaller than 1.72 were categorized as a “deletion” (−1), CNV values larger than 2.78 were categorized as a “gain” (+1), and CNVs were categorized as “normal” (0).

Cox regression (or proportional hazards regression) [102] was analyzed using the Coxph() in R. This analysis investigates the effect between the gene expression or CNVs and patient survival. The confounding factors of race and metastasis status were justified in the multivariate survival model [103].

### 4.5. Correlative Analysis between CNV and Gene Expression

Correlation analysis between gene expression and CNV was analyzed using R package, lm(). The correlation analysis was analyzed at the per-gene level.

### 4.6. Differential Expression Calculation

Differential gene expression (DEG) was analyzed by the R ‘limma’ package. The Wilcoxon rank sum test was implemented to analyze the differences between sample types. Benjamini and Hochberg adjustment was applied to all initial *p*-values, where applicable, to account for the multiple testing issues. The EdgeR [104] tool was used for RNASeq DEG calculation, which expects raw counts the as input.

### 4.7. Pathway Analysis (Gene Enrichment Analysis and Network Analysis)

GSEA software was used for pathway gene set enrichment analysis. Ingenuity Pathway Analysis (IPA, QIAGEN Redwood City, CA, USA) was used to conduct pathway analysis and identify genes connected to pathways of interest [105]. More specifically, we focused on the IPA networks that were curated from the literature.

### 4.8. Cell Lines

NHOSTs were purchased from Lonza Clonetics and expanded/grown using the Lonza Clonetics Osteoblast Growth Medium, as per the manufacturer’s instructions. Established Pediatric OS Cell Lines Saos2, G292, MG63, and U2OS were purchased from ATCC in 2018. Saos-LM7 was a kind gift from Dr. Eugenie S. Klenerman (MD Anderson, Houston, TX, USA). Pediatric OS cell lines (Saos2, Saos-LM7, G292, MG63, and U2OS) were authenticated for their identity/species by DNA fingerprinting analysis using a nine marker short-tandem repeat (STR) analysis (IDEXX BioResearch, Columbia, MO, USA), as described in [106], and were reported to be 100% human. All cell lines were cryopreserved at a low passage and were negative for mycoplasma (MycoAlert Kit; Lonza, Morristown, NJ, USA). All cell lines were cultured in DMEM containing 10% FBS. Cells were cultured at 37 ^°^C with 5% CO_2_.

### 4.9. Compounds

OTX-015 and LY2606368 (MedChem Express, Monmouth Junction, NJ, USA) and SRA737 (Chemietek, Indianapolis, IN, USA) were dissolved in 100% dimethyl sulfoxide (DMSO) for in vitro studies. The final concentration of DMSO for all drugs in all cell cultures was ≤0.1%. OTX-015 and SRA737 were dissolved in 10% DMSO, 20% PEG400, 5% Tween 80, and 65% water for the in-vivo studies.

### 4.10. Cell Proliferation Assay

OS cell lines were cultured in DMEM containing 10% FBS and seeded overnight in 96-well plates (3500 cells/well). The next day, cells were treated with varying concentrations of OTX-015, LY2606368, and SRA737 as single agents for 5 days. Following the 5-day treatment, cell growth/proliferation was determined by methylene blue staining, as previously described [106]. Experiments were conducted in sextuplicate and repeated three times.

### 4.11. In Vitro Analysis of Apoptosis by Activated Caspase-3/7

Saos2 cells (6000 cells/well) were seeded in a flat-bottom 96-well plate to adhere overnight. The following day, Saos2 cells were treated with appropriate BETi/OTX-015, CHK1i/SRA737, and vehicle concentrations. Treatments was left on the cells for 2 days without the replacement of media. Following the 2 day treatment exposure, the ApoTox-Glo Triplex Assay (Promega, Madison, WI, USA) was used as per the manufacturer’s instructions to evaluate the levels of activated caspase-3/7 (a marker for apoptosis) by reading luminescence values. Experiments were conducted in sextuplicate and repeated twice.

### 4.12. Apoptosis Flow Cytometry

Saos-2 cells were treated with a vehicle (DMSO) or 0.31 µM BETi/OTX-015 and 3.75 µM CHK1i/SRA737, as single agents or in combination. After 96 h, media plus cells were collected, placed in a tube, and washed twice with cold PBS, and 120 uL Annexin V Binding Buffer was added to the cells. Each cell was stained with 5 uL FITC Annexin V (BD Pharmingen, cat# 556420; Franklin, NJ, USA) and Propidium Iodide Staining Solution (Invitrogen, cat# 00699050; Grand Island, NY, USA) for 15 min at room temperature in the dark. Subsequently, 400 uL of 1:10 diluted Annexin V Binding Buffer (BD Pharmingen, cat# 556454) was added to each tube and stained cells were analyzed by flow cytometry on the BD Fortessa. For each sample, 20,000 cells were collected and analyzed for apoptotic cells by FlowJo Software.

### 4.13. Transient Knockdown of BRD4 with siRNA

ON-TARGET plus non-targeting control and SMARTpool ON-TARGET plus BRD4 siRNAs were purchased from Horizon Discovery. Non-targeting control siRNA sequences were D-001810-10-05: UGGUUUACAUGUCGACUAA, UGGUUUACAUGUUGUGUGA, UGGUUUACAUGUUUUCUGA, and UGGUUUACAUGUUUUCCUA. BRD4 siRNA sequences from smartpool were Pool 1, J-004937-06: AAACCGAGAUCAUGAUAGU; Pool 2, J-004937-07: CUACACGACUACUGUGACA; Pool 3, J-004937-08: AAACACAACUCAAGCAUCG; and Pool 4, J-004937-09: CAGCGAAGACUCCGAAACA. Lipofectamine RNAiMAX (Life Technologies, Grand Island, NY, USA) was used for transfection. For Western blot and proliferation assays, Saos2 cells were transfected with either 100 nM non-targeting control siRNA or 100 nM BRD4 siRNA. For Western blot, cells were seeded overnight in 10 cm Petri dishes. The next day, cells were transfected with either non-targeting control siRNA or BRD4 siRNA and subsequently collected at 24, 48, and 72 h post-transfection to assess BRD4 knock-down. For proliferation assays, Saos2 cells were seeded in 96-well plates overnight, and 1 day post-transfection, the cells were treated with the vehicle, or CHK1i/SRA737 alone or in combination for 5 days to assess cell-growth inhibition.

### 4.14. NOD.Cg-Prkdcscid Il2rgtm1Wjl/SzJ (NSG)

NSG mice were obtained from the on-site breeding colony from the In Vivo Therapeutics Core of the Indiana University Simon Comprehensive Cancer Center. All procedures were approved by the Institutional Animal Care and Use Committee at the IU School of Medicine (IACUC study # 19052). Animals were maintained under pathogen-free conditions and maintained on a Teklad Lab Animal Diet (TD 2014, Harlan Laboratories USA) with ad libitum access to sterile tap water under a 12-h light-dark cycle at 22–24 °C.

### 4.15. Development of an Saos2 CDX Model of OS

We found that it was challenging to generate sufficient numbers of mice for in vivo studies when xenografts were established by the direct implantation of Saos2 cell suspensions. While tumors grew, the cell growth kinetics were inconsistent between animals. Therefore, to increase the consistency of tumor take and growth kinetics, a two-step approach was utilized. Saos2 CDX were first established in female NSG (6–8 week old mice) by the implantation of 5 × 10^6^ Saos2 cells (200 µL 1:1 Matrigel to basal RPMI). We found that matching the gender of the tumor sample with the gender of the NSG mice (6–8 weeks old) used for tumor implant improves tumor take and allows for more consistent tumor growth kinetics. Once tumor volumes reached ~1000 mm^3^, Saos2 tumors were harvested and similar-sized tumor fragments (2 × 2 mm) were implanted into the flank of female NSG mice. Growth kinetics were tracked following implantation by an electronic caliper and implanted mice that showed consistent tumor take and growth kinetics were included in the study. In the re-implanted mice, as soon as the tumor volumes were 100–200 mm^3^, mice were randomized and treated with four 5-day cycles of SRA737, OTX-015, or combination therapy, as listed above in Figure 8.

### 4.16. Development of TT2-77 PDX from a Pediatric Patient with Relapsed OS

TT2-77 is an OS sample derived from a male pediatric OS patient that relapsed in August 2017 and was enrolled in the Precision Genomics Program at Riley Children’s Hospital (Indiana Pediatric Biobank IRB # 1501467439). The PDX was developed as described by Mattar et al. [107]. Briefly, the tumor specimen was cut into fragments and either flash frozen (5 × 5 mm)/cryopreserved (3 × 3 mm) to a biobank, or fragment (2 × 2 mm) implanted into male NSG mice. The TT2-77 PDX tumors were expanded into larger cohorts of male mice for PDX archiving and for in vivo investigations in this study. Additionally, TT2-77 PDXs were authenticated by STR analysis and MYC-RAD21 copy numbers were obtained from whole genome sequencing (Appendix A). For in vivo study set up (Figure 9), once mice tumor volumes reached at least 1000 mm^3^, the TT2-77 PDX tumor was fragmented. Similar-sized tumor fragments (2 × 2 mm) were implanted into the flank of male NSG mice. Growth kinetics were tracked following implantation by an electronic caliper and implanted mice that showed consistent tumor take and growth kinetics were included in the study. In the re-implanted mice, as soon as the tumor volumes were 100–200 mm^3^, mice were randomized and treated with four 7-day cycles of SRA737, OTX-015, or combination therapy, as described in Figure 9.

### 4.17. Development of TT2-77 Xenoline from PDX

Established TT2 tumors from the PDX model described below were resected, collected (<2000 mm^3^), and processed for single-cell separation to culture in vitro. Primary PDX tumor tissue was dissociated into a single-cell suspension by mechanical dissociation using the Gentle MACS Dissociator (Miltenyi Biotec, Gaithersburg, MD, USA) and enzymatic degradation of the extracellular matrix using the Human Tumor Dissociation Kit (Miltenyi Biotec; according to the manufacturer’s protocol). The cell suspension was washed and centrifuged three times before the cell pellet was re-suspended in DMEM or DMEM/F12 medium containing 2.5% FBS and 2 mM L-glutamine. The cell suspension was also filtered through a BD-Falcon 70-μm cell strainer. Both filtered and non-filtered cells were cultured in DMEM or DMEM/F12 medium containing 2.5% FBS and 2 mM L-glutamine individually, cultured on both Corning collagen-coated and non-coated plates or T75 flasks, and stored in a humid atmosphere of (Corning, Corning, NY, USA) at 37 °C plus 5% CO_2_. Half of the culture medium was changed every 2–3 days and the culture plate was routinely checked for adherent cells. This xenoline was generated at our institution. Furthermore, the xenoline was authenticated by STR and confirmed to be negative for mycoplasma, as described above. MYC-RAD21 copy numbers were obtained from whole genome sequencing (Appendix A).

### 4.18. Western Blot Analysis

Cells were lysed with RIPA buffer containing EDTA-free Complete Protease Inhibitor Cocktail as per the manufacturer’s instructions (Roche Diagnostics, Indianapolis, IN, USA), and 1% phosphatase inhibitor 3 as previously described Sigma [25,46]. The protein concentration was determined by the *DC* Protein Assay (Bio-Rad, Hercules, CA, USA) and quantified using BioTek Synergy H4 (BioTek, Winooski, VT, USA)). For Western blot analysis, proteins were separated on TGX Stain-Free^TM^ gels (Bio-Rad), along with Precision Plus Protein^TM^ All-Blue Standards (Bio-Rad), and transferred to an LF PVDF membrane using the Trans-Blot Turbo Transfer System (Bio-Rad). Membranes were blocked for 1 h at room temperature in 5% non-fat dry milk in TBS-T (137 mM NaCl, 20 mM Tris, and 0.02% Tween 20). Membranes were washed following antibody incubation with TBS-T for a total of three 10-min washes. The correct molecular weight for each protein was confirmed by the Precision Plus All Blue Standard (Bio-Rad). The following antibodies were diluted in either 5% non-fat dry milk or 5% BSA in TBS-T, as per the manufacturer’s instructions, and used for detection: Rabbit anti-phospho-CHK1 serine 345 [pCHK1-Ser345] (56 kDa, cat# 2348, Cell Signaling Technology, Danvers, MA, USA); mouse IgG1 anti-CHK1 [2G1D5] (56 kDa, cat# 2360, Cell Signaling Technology); mouse anti-c-Myc [9E10] (57–65 kDa, cat# MA1-980, ThermoFisher); rabbit anti-BRD2 [D89B4] (110 kDa, cat# 5848, Cell Signaling Technology); mouse anti-BRD3 [2088C3a] (80 kDa, cat# sc-81202, Santa Cruz Biotechnology, Dallas, TX, USA); rabbit anti-BRD4 [E2A7X] (200 kDa, cat# 13440, Cell Signaling Technology); rabbit anti-RAD21 (130 kDa, cat# 4321, Cell Signaling Technology); rabbit anti-phospho-H2A.X serine 139 [p-H2AX-Ser139] (15 kDa, cat# 2577, Cell Signaling Technology); rabbit anti-H2A.X (15 kDa, cat# 2595, Cell Signaling Technology); rabbit anti-cleaved PARP [D64E10] (89 kDa, cat# 5625, Cell Signaling Technology); rabbit anti-PARP (89 and 116 kDa, cat# 9542, Cell Signaling Technology); and rabbit anti-vinculin [E1E9V] (124 kDa, cat# 13901, Cell Signaling Technology). Blots were then incubated with the appropriate horseradish peroxidase-conjugated secondary antibody, diluted 1:5000 in 5% non-fat dry milk for 1 h at room temperature (Anti-mouse IgG HRP-conjugate, cat# W4021, Promega; Anti-rabbit IgG HRP-conjugate, cat# W4011, Promega). Membranes were again washed following secondary antibody incubation with TBS-T for a total of three 10-min washes. Proteins were detected using the SuperSignal Western Chemiluminescent Substrate (Thermo Scientific, Grand Island, NY, USA) and imaged using the Bio-Rad ChemiDoc Imaging System (Bio-Rad). The quantification of protein expression was done using the Image Lab software (Bio-Rad) and proteins of interest were normalized to total protein, as quantitated from the corresponding blot, and expressed relative to control cells (NHOSTs), media- or vehicle-treated control cells, or non-targeting control siRNA. For phosphorylated CHK1 and H2AX, protein levels were also quantitated and normalized to their total protein levels, respectively.

### 4.19. Statistical Analysis

For in vitro drug combination studies, IC50 values and combination indices were determined by CalcuSyn v2 (BioSoft), as previously described [106]. A Bliss independence model was used to evaluate combination effects on an efficacy scale [108]. The Bliss expected value was calculated using the equation (A + B) – (A × B), in which A and B are the percentage of growth inhibition induced by agents A and B for a given pair of doses. Values of the difference between the Bliss expected growth inhibition and the observed growth inhibition of approximately 0% indicate additivity, >0% indicate synergy, and <0% indicate antagonism. Results are reported as mean ± SD. Data were considered significant at *p* < 0.05. Saos2 flank tumor xenograft and TT2-77 PDX tumor volume data were analyzed by a two-way repeated measures ANOVA (one factor repetition), followed by a Holm–Sidak post-hoc pairwise multiple comparisons test using GraphPad Prism Software (GraphPad, Inc., San Diego, CA, USA) and SigmaPlot 11.2 (Systat Software, Inc., San Jose, CA, USA). The tumor volume is presented as the mean ± SEM and was determined 3× per week. Saos2 CDX and TT2-77 PDX survival was assessed by Kaplan–Meier plots and analyzed by the log-rank test.

## 5. Conclusions

Discoveries made possible by systems biology approaches provide innovative and rational approaches for exploring and prioritizing new possibilities for pediatric cancer treatment. Through an integrative bioinformatic analysis of CNVs, gene expression, patient outcome data, and network analysis, we identified MYC-RAD21+ as a prognostic signature that is predictive of poor overall survival in pediatric and AYA OS. This approach demonstrates the feasibility of using a CNV to identify a dysregulated network that can be therapeutically targeted at multiple points of regulation. Our data indicate that dual-inhibition of BET + CHK1 was safe and efficacious in preclinical pediatric and AYA OS models harboring the MYC-RAD21+ signature. The integration and implementation of this type of precision genomics-based methodology will help to prioritize the testing of innovative treatments to target therapeutically actionable prognostic signatures at the time of diagnosis for improving clinical outcomes in pediatric and AYA OS patients.

## Figures and Tables

**Figure 1 cancers-12-02426-f001:**
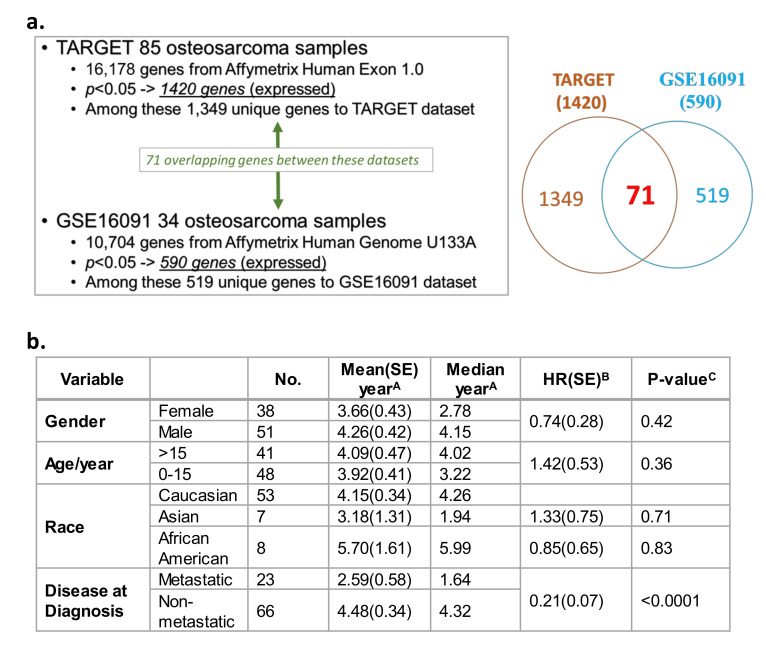
Transcriptome and survival analysis in the two independent overall survival (OS) datasets Therapeutically Applicable Research to Generate Effective Treatments (TARGET) and GSE16091. (**a**) Correlation of gene expression profiles, and OS patient overall survival in two independent datasets: TARGET and Gene Expression Omnibus (GEO) GSE16091. (**b**) Survival analysis based on demographic and clinical variables (gender, age, race, and metastasis) present within the TARGET data. Standard error, SE; hazard ratio, HR. ^A^ Survival was defined as time in years between diagnosis and death or study endpoint. ^B^ HR was calculated by a Cox regression model using the characteristic with a blank HR value as the baseline. ^C^ Relationship between a variable and survival was evaluated using a log-rank test.

**Figure 2 cancers-12-02426-f002:**
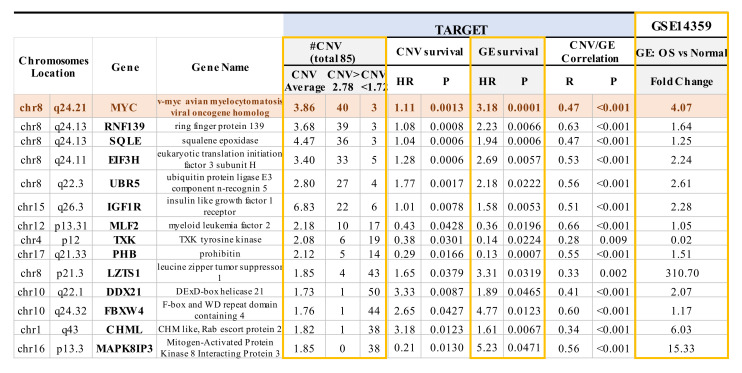
Significant correlations between copy number variation (CNV) and OS patient survival, gene expression and OS patient survival, and CNV and gene expression were evident in 14 genes. The expression of these 14 genes across various chromosomes from TARGET was also compared to non-neoplastic primary osteoblasts from GSE14359. CNVs in 14 genes are associated with the overall survival of OS based on TARGET and GSE16091 datasets. HR > 1 between overall survival and gene expression or CNVs was observed in 10 out of 14 genes, suggesting that copy number gain or increased gene expression results in an increased risk for poor overall survival. CNVs > 2.78 indicates copy number gain, whereas CNVs < 1.72 suggests loss/deletion. HR < 1 between overall survival and gene expression or CNVs was observed in 4 out of 14 genes, suggesting that copy number deletion or decreased gene expression results in an increased risk for poor overall survival. Notably, HR = 1 means that neither gene expression nor CNVs have an effect on the overall survival of OS patients. GE, gene expression.

**Figure 3 cancers-12-02426-f003:**
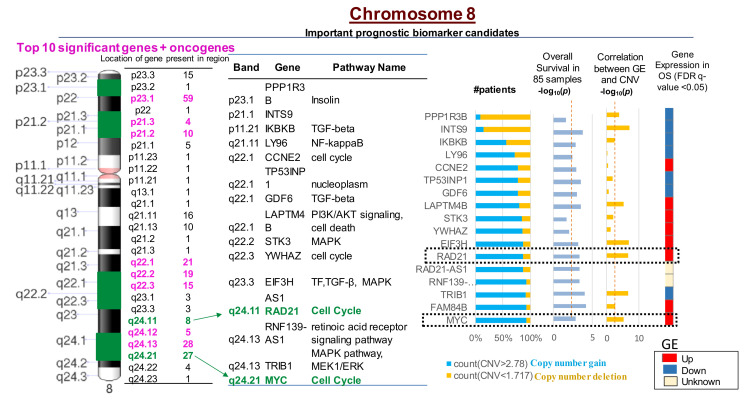
Most significant CNVs correlated with poor overall survival in OS patients are present on chromosome 8. There are 260 genes with CNVs present on chromosome 8 that are correlated with increased gene expression (GE) and poor survival in OS. The MYC-RAD21+ signature on chromosome 8q is among the top 10 most significant copy number gains correlated with poor OS survival in over 90% of patients (*p* < 0.001) and increased GE (*p* < 0.0001). CNV < 1.72 indicates deletions; CNV > 2.78 suggests copy number gains. The red dotted line in the columns (i.e., overall survival and correlation between GE and CNV -log_10_ (*p*) values) indicates statistically significant differences between the variables, *p* < 0.05.

**Figure 4 cancers-12-02426-f004:**
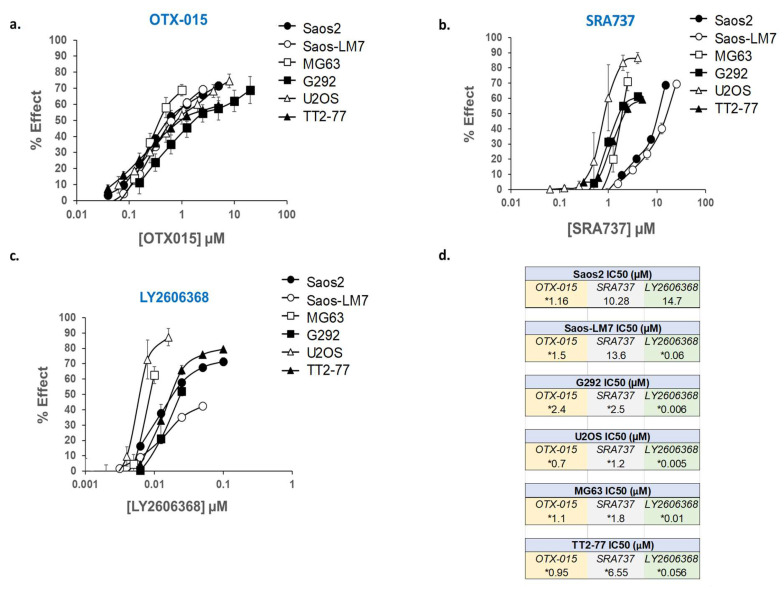
Dose-response curves for OTX-015, SRA737, and LY2606368, evaluated as single agents in a panel of established pediatric OS cell lines and TT2-77 xenoline. Dose-related inhibition of cell growth in the presence of (**a**) OTX-015, (**b**) SRA737, and (**c**) LY2606368 in pediatric OS cell lines. (**d**) Concentration required as single agents to inhibit 50% cell growth (IC50) in pediatric OS cell lines. Cell proliferation was determined by methylene blue staining after 5 days of drug exposure. Each point represents the mean of three experiments, each conducted in triplicate. Vertical lines represent ± one standard deviation and are absent when less than the size of the point. µM, micromolar. * Within the range of clinically achievable concentrations in human plasma.

**Figure 5 cancers-12-02426-f005:**
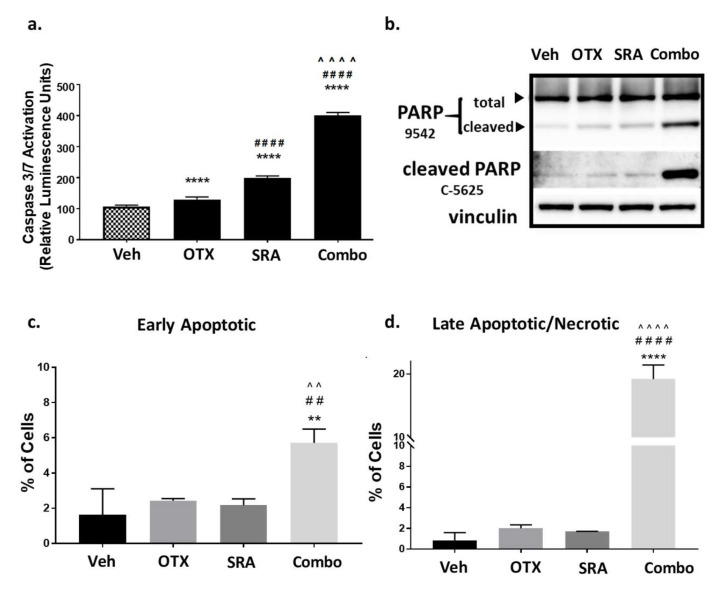
Combination BETi/OTX-015 + CHK1i/SRA737 increases apoptotic cell death in OS cells. (**a**–**b**) Saos2 cells were treated with a vehicle, BETi/OTX-015, CHK1i/SRA737, or combination of BETi/OTX-015 + CHK1i/SRA737 for 2 days and analyzed for apoptosis by (**a**) caspase 3/7 activation via the Apotox assay, and (**b**) PARP cleavage by Western blot. (**c**) Saos2 cells were treated with the inhibitor concentrations noted above for 4 days. Apoptosis was analyzed by Annexin V/Propidium Iodide staining and flow cytometry to evaluate (**c**) early apoptotic (AV+, PI−) and (**d**) late apoptotic/necrotic (AV+, PI+) cells. For the Caspase-3/7 Apotox assay, there were *n* = 6 replicates per group and data are representative of two independent experiments. The PARP Western blot was performed once. The apoptosis flow data had *n* = 3 technical replicates and is representative of two independent experiments. ** *p* < 0.01 vs. vehicle; ## *p* < 0.01 vs. BETi/OTX-015; ^^ *p* < 0.01 vs. CHK1i/SRA737; **** *p* < 0.0001 vs. vehicle; #### *p* < 0.0001 vs. BETi/OTX-015; ^^^^ *p* < 0.0001 vs. CHK1i/SRA737; data analyzed by one-way ANOVA followed by a Holm–Sidak post-hoc pairwise multiple comparisons test. Veh = vehicle; OTX = BETi/OTX-015 (0.31 µM); SRA = CHK1i/SRA737 (3.75 µM); Combo = OTX (0.31 µM) + SRA (3.75 µM).

**Figure 6 cancers-12-02426-f006:**
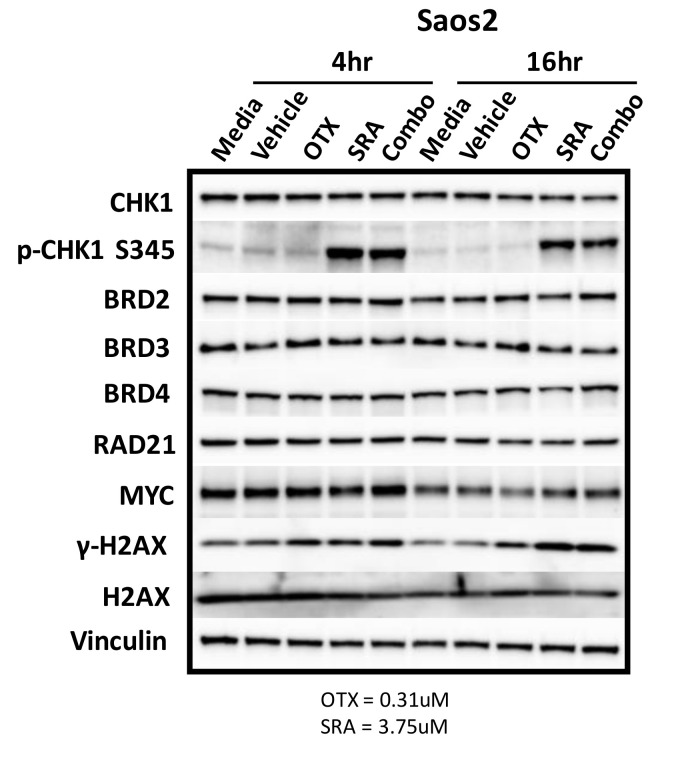
BET inhibition as a single agent or in combination with CHK1i/SRA737 does not decrease the MYC protein. Saos2 cells were treated with controls (media and vehicle), as well as single-agent and combination therapy involving BETi/OTX-015 (0.31 µM) and/or CHK1i/SRA737 (3.75 µM) for 4 or 16 h. Concentrations were obtained from data in Table 2a. Vinculin serves as the loading control. This was performed once. *OTX = BETi/OTX-015* and *SRA = CHK1i/SRA737.*

**Figure 7 cancers-12-02426-f007:**
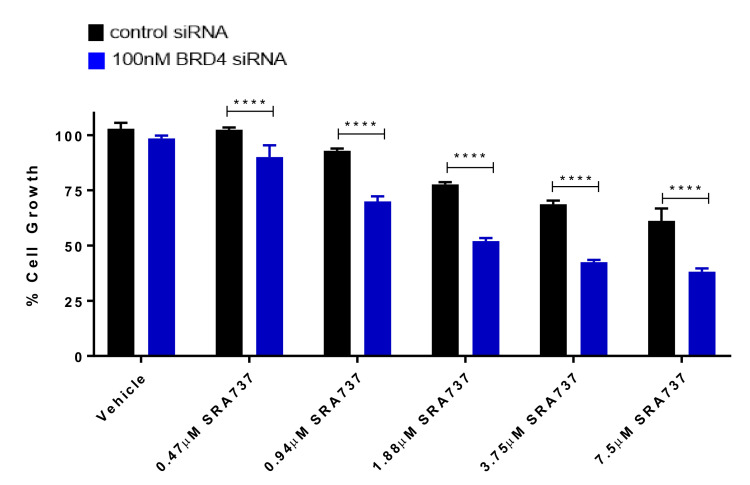
BRD4 knockdown enhances CHK1i/SRA737-mediated growth inhibition. Following transfection for 24 h (control siRNA or 100 nM BRD4 siRNA pools), Saos2 cells were treated with CHK1i/SRA737 for 5 days and cell growth was evaluated by methylene blue staining. One-way ANOVA followed by a Holm–Sidak post-hoc pairwise multiple comparisons test, *n* = 3; **** *p* < 0.0001 vs. control siRNA. Data are representative of two independent experiments.

**Figure 8 cancers-12-02426-f008:**
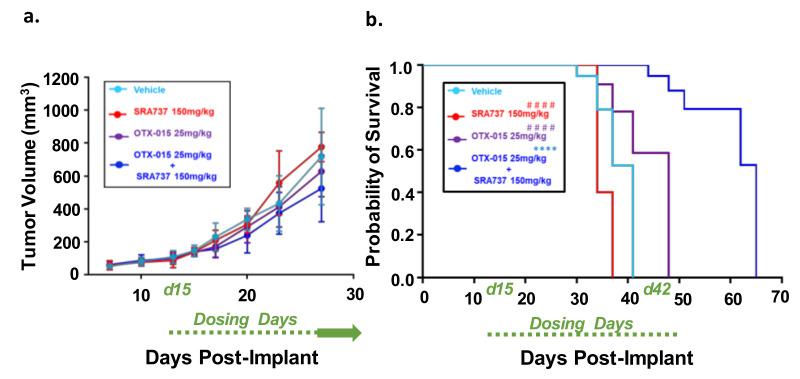
Combination therapy targeting BET and CHK1 increases the survival of mice with MYC-RAD21+ Saos2 CDX. (**a**) Tumor volumes are shown while all mice are still under study. Once tumor xenografts reached 100–200 mm^3^, mice were randomized and treated with four 5-day cycles of BETi/OTX-015 (25 mg/kg BID, days 1 through 5 of each cycle) and CHK1i/SRA737 (150 mg/kg SID, days 2 and 4 of each cycle) as single agents or as a combination, or the respective vehicle(s). There were *n* = 7 mice for the vehicle group and *n* = 4–5 mice for treatment groups. For cycles 1 and 2 of treatment, SRA737 was given 2×/week and was well-tolerated. As previously mentioned, Saos2 is relatively resistant to SRA737 compared to other OS cell lines (Figure 4). Therefore, since the drug was well-tolerated in cycles 1 and 2, SRA737 doses were increased to 3×/week in single-agent and combination treatment groups in cycles 3 and 4. Data represent tumor volumes when all mice were still under study. (**b**) Kaplan–Meier survival plot. An endpoint tumor volume of 1500 mm^3^ was used for analysis. **** *p* < 0.0001 vs. vehicle; #### *p* < 0.0001 vs. BETi/OTX-015 + CHK1i/SRA737. Dosing period: Day 15-day 42 post-implant (d15–d42).

**Figure 9 cancers-12-02426-f009:**
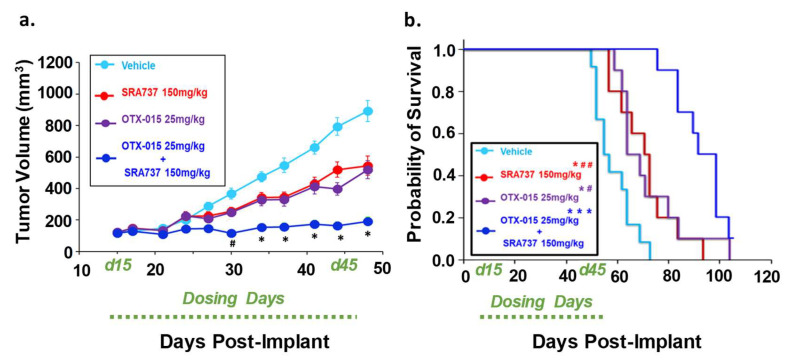
Combination therapy targeting BET and CHK1 significantly inhibited initial tumor growth and increased the survival of mice with a pediatric relapsed MYC-RAD21+ PDX. (**a**) Tumor volumes are shown while all mice are still under study. Once tumor volumes were 100–200 mm^3^, mice were randomized (*n* = 8–10 per group) and treated with four 5-day cycles of BETi/OTX-015 (25 mg/kg BID, days 1 through 5 of each cycle) and CHK1i/SRA737 (150 mg/kg SID, days 2 and 4 of each cycle) as single agents or as a combination, or the respective vehicle(s). # *p* < 0.05, combination (BETi/OTX-015 + CHK1i/SRA737) vs. vehicle or single agents. * *p* < 0.001 combination (BETi/OTX-015 + CHK1i/SRA737) vs. vehicle or single agents through day 48. (**b**) Kaplan–Meier survival plot. An endpoint tumor volume of 1500 mm^3^ was used for analysis. * *p* < 0.05; *** *p* < 0.001 vs. vehicle; # *p* < 0.05; ## *p* < 0.01. Of note: The tumor volume of one mouse in the *BETi/OTX-015 + CHK1i/SRA737* combination group was still <1500 mm^3^ on the last day of the study (Day 100), so was censored. Dosing period: Day 15-day 45 post-implant (d15*–*d45).

**Figure 10 cancers-12-02426-f010:**
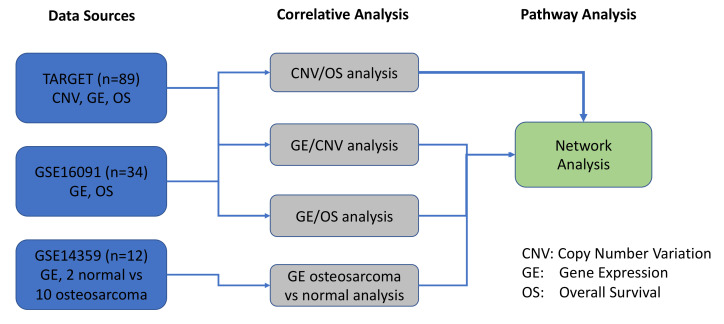
Schematic overview of the datasets and bioinformatics analyses conducted in the study.

**Table 1 cancers-12-02426-t001:** Enrichment analysis of significant CNVs observed in genes across various chromosomes. The enrichment odds-ratio (OR) of a chromosome means the ratio of the CNV gene-enriched odds from this chromosome over the other chromosomes’ CNV gene-enriched odds. The enrichment *p*-value was calculated from a chi-square test from a 2 by 2 table, in which CNV gene enrichment in one chromosome was compared to all the others. The *p*-value was obtained from a one-sided test for OR > 1. Chr, chromosome. Chromosomes listed in red have significantly more genes with CNVS associated with patient overall survival.

Chr	Total Gene #	Significant Gene #	Significant Gene %	CNV Gains #	CNV Loss#	*p*-Value	Enrichment Odds Ratio	−log *p*
1	2652	220	8.30	58	205	0.997	0.800	0.003
2	1728	56	3.24	16	56	1.000	0.287	0.000
3	1477	60	4.06	18	60	1.000	0.368	0.000
4	1011	88	8.70	47	82	0.813	0.856	0.207
5	1216	26	2.14	9	26	1.000	0.189	0.000
6	1374	21	1.53	15	21	1.000	0.133	0.000
7	1269	10	0.79	6	10	1.000	0.068	0.000
8	962	261	27.13	188	160	1.05 × 10^−72^	3.617	165.736
9	1063	132	12.42	41	132	7.75 × 10^−03^	1.294	4.860
10	1061	367	34.59	8	367	1.44 × 10^−163^	5.375	374.960
11	1609	428	26.60	129	428	3.13 × 10^−116^	3.705	265.958
12	1326	127	9.58	116	127	0.344	0.954	1.067
13	601	6	1.00	2	6	1.000	0.089	0.000
14	876	12	1.37	11	2	1.000	0.121	0.000
15	931	193	20.73	81	191	1.33 × 10^−28^	2.467	64.185
16	1086	280	25.78	7	278	4.18 × 10^−70^	3.386	159.750
17	1498	207	13.82	70	205	4.02 × 10^−7^	1.485	14.728
18	407	12	2.95	2	12	1.000	0.271	0.000
19	1725	108	6.26	77	108	1.000	0.586	0.000
20	749	10	1.34	6	10	1.000	0.119	0.000
21	341	1	0.29	1	1	1.000	0.026	0.000
22	587	37	6.30	35	37	0.997	0.602	0.003
X	1118	8	0.72	7	8	1.000	0.062	0.000
Y	106	0	0.00	0	0	0.999	0.000	0.001

**Table 2 cancers-12-02426-t002:** Analysis of cell growth inhibition in pediatric OS cell lines treated with BET and CHK1 inhibitors. Established OS cells and TT2-77 OS xenoline were treated with varying dose ratios of BETi/OTX-015 and (**a**) CHK1i/SRA737 or (**b**) CHK1i/LY2606368. Cell growth was determined by methylene blue staining. Efficacy synergy was determined by Bliss Independence analysis. SEM = standard error mean. * Within the range of clinically achievable concentrations. Clinically achievable doses include BETi/OTX-015 (1–2 µM) [58], CHK1i/SRA737 (6 µM) [63], and CHK1i/LY2606368 (0.1–0.2 µM) [74]. Data are compiled from three independent experiments.

(a)									
**Saos2: Mean+SEM Observed–Expected (% Effect)**									
**OTX-015 (µM)**									
**SRA737 (µM)**		***0.04**	***0.08**	***0.16**	***0.31**	***0.63**	***1.25**	**2.5**	**5**
	***0.94**	3 ± 0	11 ± 1	13 ± 1	16 ± 0	15 ± 0	16 ± 0	15 ± 0	15 ± 1
	***1.88**	9 ± 1	16 ± 1	20 ± 1	20 ± 1	19 ± 0	20 ± 1	17 ± 0	15 ± 0
	***3.75**	10 ± 0	17 ± 0	19 ± 1	21 ± 0	19 ± 1	17 ± 1	14 ± 0	13 ± 0
	**7.5**	7 ± 0	12 ± 1	15 ± 1	16 ± 0	15 ± 1	13 ± 1	12 ± 0	9 ± 1
	**15**	−3 ± 1	−1 ± 0	0 ± 1	3 ± 0	3 ± 0	2 ± 1	1 ± 1	−1 ± 0
**Saos-LM7: Mean ± SEM Observed–Expected (% Effect)**									
**OTX-015 (µM)**									
**SRA737 (µM)**		***0.02**	***0.04**	***0.08**	***0.16**	***0.31**	***0.63**	***1.25**	**2.5**
	***1.56**	6 ± 1	3 ± 2	4 ± 2	7 ± 2	9 ± 2	10 ± 2	8 ± 1	7 ± 1
	***3.13**	9 ± 0	7 ± 1	8 ± 3	7 ± 2	9 ± 2	12 ± 1	8 ± 1	6 ± 1
	**6.25**	6 ± 2	5 ± 2	7 ± 1	6 ± 1	4 ± 1	4 ± 2	4 ± 1	2 ± 1
	**12.5**	2 ± 1	2 ± 1	4 ± 0	4 ± 1	4 ± 0	3 ± 1	3 ± 1	0 ± 1
	**25**	2 ± 0	2 ± 1	0 ± 3	5 ± 1	4 ± 1	3 ± 1	0 ± 1	−2 ± 1
**MG63: Mean ± SEM Observed–Expected (% Effect)**									
**OTX-015 (µM)**									
**SRA737 (µM)**		***0.02**	***0.04**	***0.08**	***0.16**	***0.31**	***0.63**	***1.25**	**2.5**
	***0.16**	−7 ± 1	−7 ± 1	−6 ± 1	−6 ± 0	−2 ± 0	0 ± 1	1 ± 0	−1 ± 0
	***0.31**	−9 ± 0	−10 ± 1	−7 ± 1	−6 ± 0	−4 ± 0	0 ± 1	1 ± 1	0 ± 0
	***0.63**	−7 ± 1	−8 ± 1	−6 ± 1	−6 ± 0	−1 ± 0	3 ± 1	6 ± 0	4 ± 1
	***1.25**	−6 ± 0	−11 ± 1	−14 ± 2	−14 ± 3	−9 ± 3	0 ± 2	3 ± 3	2 ± 1
	***2.5**	−6 ± 1	−17 ± 2	−30 ± 2	−36 ± 1	−30 ± 1	−18 ± 1	−14 ± 1	−13 ± 1
**G292: Mean ± SEM Observed–Expected (% Effect)**									
**OTX-015 (µM)**									
**SRA737 (µM)**		***0.13**	***0.25**	***0.5**	***1**	***2**	***4**	***8**	**16**
	***0.25**	−7 ± 0	−10 ± 0	−7 ± 0	−9 ± 1	−4 ± 0	−5 ± 2	−3 ± 1	0 ± 1
	***0.5**	−5 ± 2	−8 ± 1	−7 ± 1	−8 ± 1	−3 ± 1	−2 ± 1	0 ± 1	2 ± 2
	***1**	−13 ± 3	−18 ± 2	−17 ± 2	−17 ± 2	−11 ± 1	−10 ± 2	−7 ± 2	−3 ± 2
	***2**	−16 ± 2	−22 ± 1	−24 ± 2	−22 ± 2	−19 ± 2	−16 ± 2	−13 ± 1	−8 ± 1
	***4**	−16 ± 2	−22 ± 2	−25 ± 1	−22 ± 2	−19 ± 2	−16 ± 2	−14 ± 2	−9 ± 2
**U2OS: Mean ± SEM Observed–Expected (% Effect)**									
**OTX-015 (µM)**									
**SRA737 (µM)**		***0.02**	***0.05**	***0.09**	***0.18**	***0.38**	***0.75**	***1.5**	**3**
	***0.09**	1 ± 1	−1 ± 2	0 ± 1	1 ± 1	2 ± 1	0 ± 0	1 ± 0	1 ± 1
	***0.19**	0 ± 2	−3 ± 2	0 ± 2	3 ± 0	2 ± 1	2 ± 0	3 ± 1	2 ± 1
	***0.38**	−8 ± 2	−11 ± 1	−10 ± 1	−7 ± 3	−6 ± 3	−5 ± 3	−4 ± 3	−3 ± 3
	***0.75**	−8 ± 3	−9 ± 3	−10 ± 4	−10 ± 2	−10 ± 2	−10 ± 1	−9 ± 0	−9 ± 1
	***1.5**	2+1	−4 ± 1	−6 ± 1	−8 ± 1	−8 ± 1	−9 ± 1	−9 ± 1	−8 ± 0
**TT2-77 Xenoline: Mean ± SEM Observed–Expected (% Effect)**									
**OTX-015 (µM)**									
**SRA737 (µM)**		***0.04**	***0.08**	***0.16**	***0.31**	***0.63**	***1.25**	**2.5**	**5**
	***0.31**	−7 ± 1	−7 ± 1	−9 ± 1	−7 ± 1	−8 ± 1	−5 ± 1	−4 ± 1	−3 ± 1
	***0.63**	−7 ± 0	−7 ± 1	−9 ± 1	−8 ± 1	−8 ± 1	−5 ± 1	−4 ± 1	−4 ± 1
	***1.25**	−9 ± 1	−8 ± 0	−11 ± 1	−10 ± 1	−10 ± 1	−6 ± 1	−6 ± 1	−5 ± 1
	***2.5**	−17 ± 2	−20 ± 4	−23 ± 4	−21 ± 4	−19 ± 4	−16 ± 3	−16 ± 3	−16 ± 3
	***5**	−17 ± 0	−21 ± 1	−25 ± 2	−25 ± 2	−24 ± 3	−21 ± 3	−21 ± 3	−21 ± 3
**(b)**									
**Saos2: Mean ± SEM Observed–Expected (% Effect)**									
**OTX-015 (µM)**									
**LY2606368 (µM)**		***0.01**	***0.02**	***0.03**	***0.06**	***0.13**	***0.25**	***0.5**	*1
	***0.00625**	1 ± 1	2 ± 1	8 ± 1	9 ± 2	16 ± 1	19 ± 1	21 ± 2	18 ± 1
	***0.0125**	4 ± 1	6 ± 2	8 ± 1	13 ± 2	17 ± 0	18 ± 0	17 ± 0	15 ± 1
	***0.025**	1 ± 1	4 ± 0	6 ± 1	9 ± 1	11 ± 0	10 ± 0	10 ± 0	6 ± 1
	***0.05**	2 ± 0	2 ± 1	4 ± 1	6 ± 0	6 ± 1	6 ± 0	5 ± 0	3 ± 1
	***0.1**	0 ± 1	1 ± 1	2 ± 1	4 ± 1	4 ± 0	4 ± 0	2 ± 0	0 ± 1
**Saos-LM7: Mean ± SEM Observed–Expected (% Effect)**									
**OTX-015 (µM)**									
**LY2606368 (µM)**		***0.01**	***0.02**	***0.03**	***0.06**	***0.13**	***0.25**	***0.5**	***1**
	***0.00313**	3 ± 1	2 ± 0	−1 ± 1	−2 ± 1	−1 ± 1	−3 ± 1	−1 ± 0	0 ± 1
	***0.00625**	4 ± 0	3 ± 1	1 ± 1	2 ± 0	0 ± 3	2 ± 1	2 ± 1	1 ± 1
	***0.0125**	7 ± 1	6 ± 1	4 ± 1	6 ± 2	8 ± 1	8 ± 1	7 ± 1	4 ± 1
	***0.025**	4 ± 1	3 ± 2	3 ± 1	6 ± 1	7 ± 1	9 ± 1	7 ± 2	4 ± 2
	***0.05**	14 ± 9	14 ± 9	13 ± 7	14 ± 9	16 ± 9	16 ± 7	12left6	8 ± 5
**MG63: Mean ± SEM Observed–Expected (% Effect)**									
**OTX-015 (µM)**									
**LY2606368 (µM)**		***0.01**	***0.02**	***0.03**	***0.06**	***0.13**	***0.25**	***0.5**	***1**
	***0.00063**	2 ± 1	2 ± 0	1 ± 1	2 ± 1	−1 ± 1	0 ± 1	0 ± 2	2 ± 1
	***0.00125**	2 ± 1	2 ± 2	1 ± 1	2 ± 0	2 ± 1	1 ± 0	0 ± 0	0 ± 1
	***0.0025**	3 ± 2	3 ± 3	5 ± 3	2 ± 2	2 ± 1	4 ± 1	1 ± 1	2 ± 1
	***0.005**	5 ± 2	4 ± 2	2 ± 2	1 ± 2	1 ± 1	0 ± 1	1 ± 2	2 ± 1
	***0.01**	−4 ± 0	−7 ± 1	−17 ± 1	−29 ± 2	−31 ± 1	−21 ± 2	−13 ± 1	−10 ± 1
**G292: Mean ± SEM Observed–Expected (% Effect)**									
**OTX-015 (µM)**									
**LY2606368 (µM)**		***0.16**	***0.31**	***0.63**	***1.25**	**2.5**	**5**	**10**	**20**
	***0.25**	1 ± 2	−1 ± 1	1 ± 1	1 ± 1	−1 ± 0	0 ± 1	2 ± 1	2 ± 1
	***0.5**	1 ± 3	−3 ± 2	−1 ± 2	1 ± 1	−1 ± 1	1 ± 2	3 ± 1	5 ± 2
	**1**	−5 ± 4	−4 ± 3	−3 ± 2	0 ± 2	0 ± 1	2 ± 1	4 ± 1	8 ± 2
	**2**	−11 ± 3	−13 ± 2	−10 ± 2	−7 ± 2	−5 ± 1	−1 ± 2	2 ± 1	5 ± 1
	**4**	−22 ± 3	−24 ± 3	−19 ± 1	−16 ± 1	−13 ± 0	−9 ± 1	−6 ± 1	−1 ± 0
**U2OS: Mean ± SEM Observed–Expected (% Effect)**									
**OTX-015 (µM)**									
**LY2606368 (µM)**		***0.06**	***0.13**	***0.25**	***0.5**	***1**	***2**	**4**	**8**
	***0.00156**	2 ± 5	3 ± 5	2 ± 2	−1 ± 2	−1 ± 2	1 ± 3	1 ± 4	2 ± 3
	***0.003**	1 ± 2	5 ± 3	3 ± 3	0 ± 1	1 ± 2	3 ± 2	2 ± 3	4 ± 3
	***0.006**	−3 ± 2	1 ± 1	−1 ± 4	−4 ± 4	−3 ± 2	0 ± 2	1 ± 1	4 ± 1
	***0.0125**	−10 ± 1	−13 ± 2	−15 ± 1	−18 ± 2	−17 ± 1	−14 ± 2	−9 ± 2	−6 ± 2
	***0.025**	−3 ± 0	−5 ± 1	−8 ± 2	−10 ± 2	−11 ± 2	−10 ± 2	−8 ± 2	−7 ± 1
**TT2-77 Xenoline: Mean ± SEM Observed–Expected (% Effect)**									
**OTX-015 (µM)**									
**LY2606368 (µM)**		***0.04**	***0.08**	***0.16**	***0.31**	***0.63**	***1.25**	**2.5**	**5**
	***0.00625**	−4 ± 1	−7 ± 1	−8 ± 2	−9 ± 1	−6 ± 1	−5 ± 0	−3 ± 1	−3 ± 1
	***0.0125**	−14 ± 1	−18 ± 2	−22 ± 2	−21 ± 1	−18 ± 1	−15 ± 1	−14 ± 2	−13 ± 2
	***0.025**	−15 ± 1	−21 ± 1	−28 ± 1	−30 ± 1	−28 ± 1	−26 ± 1	−25 ± 1	−24 ± 1
	***0.05**	−7 ± 1	−13 ± 0	−18 ± 0	−24 ± 1	−25 ± 1	−26 ± 1	−26 ± 1	−25 ± 1
	***0.1**	−5 ± 1	−9 ± 1	−14 ± 2	−20 ± 1	−22 ± 1	−24 ± 0	−26 ± 1	−26 ± 1
**Key**									
<−10					Antagonism				
−10 to 10					Additive				
10 to 20					Synergistic				
20 to >30					Markedly Synergistic				

The colors in the table indicate the different levels of additivity, synergism, or antagonism.

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
