# Peer review of "Systems Biology Approach Identifies Prognostic Signatures of Poor Overall Survival and Guides the Prioritization of Novel BET-CHK1 Combination Therapy for Osteosarcoma"

_cancers, 2020, doi:10.3390/cancers12092426_

Round 1
Reviewer 1 Report
The authors have presented a systems biology study on osteosarcoma to decode the prognostic signatures of poor overall survival.
I have the following concerns about this paper and I believe addressing them will improve the paper:
- Fonts in some figures are very small and one can not read them. (For example figure 5)
- My main concern is about the computational method. The authors need to mention the software they have used and if they have used their own codes, I think they need to provide them as supplemental materials.
- I expect an improved introduction mentioning why systems biology is needed to address the prognostic signatures in osteosarcoma.
Reviewer 2 Report
The manuscript by Pandya and colleagues discuss about identification of prognostic signatures of poor overall survival for osteosarcoma and use of BET and CHK1 inhibitors in combination therapy. Osteosarcoma is not a cancer clearly characterized by genetic mutations or oncogenic expressions. Due to this molecular heterogeneity, it is remarkably interesting to investigate new prognostic signatures and therapeutic targets. First, using open datasets TARGET and Gene Expression Omnibus, the authors identified in osteosarcoma a correlation between high expression of 14 genes due to copy number variations and poor overall survival. Among these genes, MYC, a well-known oncogene, and RAD21, a protein involved cell cycle pathway, are highlighted by the authors. Then, using osteosarcoma cell lines expressing MYC and RAD21, the authors tested in vitro a combination with a BET inhibitor (OTX-015) and a CHK1 inhibitor (LY2606368 or SRA7737). A synergistic effect is observed in both SaOS2 cell lines but not in the others. This association were tested in vivo with some benefits.
Minor comments
Figure 1b: This survival analysis based on demographic variables brings nothing new and seems off topic
Figure 2b: The same analyze with RAD21 should be interesting.
Figure S1a: What is exactly “normal human osteoblasts” used as control? The “materials and methods” section did not mention it. Did you differentiate mesenchymal stem cells into osteoblasts in vitro yourself?
Figure 5: Another type of analysis on apoptosis (PARP cleavage by western blot, annexin V by flow cytometry…) could be appreciated.
Table S7: Whole genome sequencing of PDX model showing an amplification of MYC and RAD2121 gene copies is interesting but that needs to be correlated with protein expression.
Major comments
The authors showed in the first part of the article that a gene amplification of MYC is a poor overall survival in osteosarcoma. This result is not surprising and aberrant copy number of myc gene is already described in osteosarcoma. The main of the study is to identify biomarkers to treat osteosarcoma with specific chemotherapies. The authors identified MYC and RAD21 as signatures but did not clearly show that drug combination with BETi and CH1i is relevant precisely in this case. In vitro this association acted in a synergistic way with only one osteosarcoma cell line, although the other cell lines are also MYC-RAD21+. It will be necessary to test an osteosarcoma cell line MYC-RAD21 negative. Moreover, it is not clear why the authors chose to associate a BET and a CHK1 inhibitor to treat MYC-RAD21+ osteosarcoma cell lines and this point needs to be clarified.
Although this work shows some interesting ideas and results, the first part does not seem to be new and is disconnected from the second part.
Due to all these major aspects, I cannot recommend this article to go further but I hope the authors find these suggestions helpful.
Reviewer 3 Report
The authors Pandya et al., in their work titled "Systems biology approach identifies prognostic signatures of poor overall survival and guides prioritization of novel BET-CHK1 combination therapy for osteosarcoma", have presented their research on understanding the population of osteosarcoma patients that portend a worse prognosis and suggest targeted base therapy to treat this specific population. The identification of the over amplification of the MYC-RAD21 genes associates with patients that have lower overall survival and the therapeutic approach to use a BET inhibitor and CHK1 inhibitor appears to have significant negative consequences on the cancer cells.
The authors should be congratulated on the presentation of their information both visually and in writing. Furthermore, the details provided in the methods, introduction, and results exceeded most published manuscripts. Lastly, the number of replicates both in terms of technical and biological should be commended.
There are only a few minor suggestions to improve the quality of the manuscript and they are enumerated below.
- The only clinical variable that was significant was the disease at diagnosis (metastatic vs non-metastatic). The authors should evaluate if the metastatic group exhibited predominantly the MYC over amplification. They could perform a non-parametric test (i.e. Fishers Exact Test) to see if the ratio of MYC-RAD21+ tumors is higher than expected in the metastatic cohort.
- In Figure 3, it is unclear what the dotted thresholds represent in the -log10(p).
- In the discussion, the authors discuss whether or not MYC-RAD21 could be the sole predictor to BETi + CHK1i. In fact, they already have evidence that it is not, because the Saos2 xenograft model did not delay tumor volume growth like the combination treatment did in the TT2-77 PDX model. I would like to see the authors expound on this some more and incorporate it into their discussion and limitations.
- In Figure S3, the figure legend states that M = 100 nM BRD4 siRNA, but in the figure they use a B.
